



# Evapotranspiration feedbacks shift annual precipitation-runoff relationships during multi-year droughts in a Mediterranean mixed rain-snow climate

Francesco Avanzi[1], Joseph Rungee[2], Tessa Maurer[1], Roger Bales[2, 1], Qin Ma[2], Steven Glaser[1], and Martha Conklin[2]

[1]Department of Civil and Environmental Engineering, University of California, Berkeley, 94720, Berkeley, California, USA
[1]Sierra Nevada Research Institute, University of California, Merced, 95343, Merced, California, USA

**Correspondence:** Francesco Avanzi (francesco.avanzi@polimi.it)

**Abstract.**

Focusing on the headwaters of the California's Feather River, we investigated how multi-year droughts affect the water balance of Mediterranean mixed rain-snow catchments. Droughts in these catchments saw a lower fraction of precipitation allocated to runoff compared to non-drought years. This shift in precipitation-runoff relationship was larger in a surface-runoff-

dominated than in a subsurface-flow-dominated catchment — 39% and 18% less runoff, respectively, for a representative precipitation amount. The performance of the PRMS hydrologic model in these catchments decreased during droughts, particularly those causing larger shifts in the annual precipitation-runoff relationship. Evapotranspiration ($ET$) was the only water-balance component for which predictive accuracy during drought vs. non-drought years was consistently different. Besides a systematic bias during all years, the model tended to relatively overestimate drought $ET$ and to underestimate non-drought $ET$. Modeling

errors for $ET$ during droughts were somewhat correlated with maximum and minimum annual temperature as well as changes in sub-surface storage ($r$ = -0.45, -0.57, and 0.23, respectively). These correlations point to the interannual response of $ET$ to climate, or climate elasticity of $ET$, as the likely driver of the observed shifts in precipitation-runoff relationship during droughts in Mediterranean mixed rain-snow regions; underestimation of this response caused increased modeling inaccuracy during droughts. Improved predictions of interannual variability of $ET$ are necessary to support water-supply management in

a warming climate and could be achieved by explicitly parametrizing feedback mechanisms across atmospheric demand for moisture, $ET$, and multi-year carryover of subsurface storage

## 1   Introduction

Droughts have a profound impact on ecosystems and societies (Kuil et al., 2019), especially because they can be more persistent

than other water risks (He et al., 2017; Rungee et al., 2018). Some examples include the Dust Bowl drought in the 1930s,



which triggered long-lasting changes in the economy and population distribution of the United States (Cook et al., 2014), the Millennium Drought in south-eastern Australia (1997-2009, see Chiew et al., 2014; Saft et al., 2015), the persistently dry conditions across the U.S. southwest in the early 2000s (Cayan et al., 2010), the large-scale European drought in the 1920s (Hanel et al., 2018), and the 2016 El-Niño drought in South Africa (Baudoin et al., 2017). Since aridity will increase

in frequency and extent under a warming climate (Cayan et al., 2010; Woodhouse et al., 2010), understanding the impact of droughts on the hydrologic budget is relevant to water supply, ecosystem services, and water security (Bales et al., 2018).

Precipitation deficit has frequently been assumed as the primary proxy to explain changes in water supply during droughts (Bales et al., 2018). While low-precipitation periods generally lead to a decrease in runoff, other factors can exacerbate or alleviate this response. Some of these factors include concurrent air temperature (Griffin and Anchukaitis, 2014; He et al.,

2017), pre-drought aridity (Saft et al., 2016b), tree mortality and evapotranspiration (Bales et al., 2018), sub-surface storage (Saft et al., 2016b; Klos et al., 2018; Rungee et al., 2018), and geology (e.g., granitic vs. volcanic bedrock, see Jefferson et al., 2008; Tague et al., 2008; Tague and Grant, 2009). Current drought indices do consider a broad spectrum of climatic variables (Shukla and Wood, 2008; Hanel et al., 2018), but catchment properties can still challenge drought-impact assessments (Bales et al., 2018). In the African Sahel, for example, a multi-decadal drought has led to a poorly understood increase in runoff

(*Sahelian paradox*, see Gal et al., 2017).

An inconsistent response of runoff to droughts may be evidence that these periods lead to changes in catchment functioning, similarly to other catchment-climate coevolution processes (Troch et al., 2015; Saft et al., 2016b). Such changes in runoff response during droughts have been observed in Australia (Saft et al., 2016b), California (Bales et al., 2018), and China (Tian et al., 2018) and have usually been quantified as statistical shifts in the precipitation-runoff relationship, i.e., an empirical

regression between annual precipitation and annual runoff (Chiew et al., 2014; Saft et al., 2015; Tian et al., 2018). While these lumped interpretations allow one to predict the occurrence of shifts based on catchment and drought characteristics (Saft et al., 2016b), the internal catchment mechanism behind them has not yet been fully clarified. Runoff is ultimately the output of a water balance, that is, $Q = P - ET - \Delta S$, where $Q$ is runoff, $P$ is precipitation, $ET$ is evapotranspiration, and $\Delta S$ is the change in sub-surface storage. In a water balance, shifts thus correspond to a different allocation of $P$ across $ET, \Delta S$, and $Q$

between drought and non-drought periods. Unraveling the interplay across water-balance components is a key to clarify the mechanisms behind shifts in precipitation-runoff relationship during droughts (Bales et al., 2018).

A water-balance perspective of droughts is essential in a Mediterranean climate, where precipitation is concentrated in winter and summers are dry (Bales et al., 2018). In these regions, water stored in the form of snow or in the regolith can support evapotranspiration during multi-year droughts and offset precipitation deficit at the expenses of runoff (Fellows and

30 Goulden, 2017; Bales et al., 2018; Klos et al., 2018; Rungee et al., 2018), a mechanism that is further exacerbated by increasing temperatures and thus increased aridity (Cayan et al., 2010). California, a region with a markedly Mediterranean climate, has seen four officially designated droughts since the 1970s (water years 1976-77, 1987-92, 2007-09, and 2012-15, see He et al., 2017). Because most of precipitation in the state falls in the north and during winter, water supply is mostly generated in mixed rain-snow, geologically and topographically complex headwaters, while water is mainly consumed in lowland regions.

Rising temperatures are threatening this equilibrium (Harpold et al., 2017; Hatchett and McEvoy, 2018), but the impact of





droughts on Mediterranean, mixed rain-snow catchments has rarely been studied from a water-balance perspective, meaning both hydrologic-model predictive accuracy and drought-management solutions are still uncertain (Bales et al., 2018).

The four Californian droughts between the 1970s and 2010s offer an opportunity to clarify the mechanisms behind shifts in precipitation-runoff relationships in a Mediterranean, mixed-rain snow climate, as well as the adequacy of hydrologic models

5  to simulate them. To achieve this goal, we used detailed water-balance indices and hydrologic modeling (PRMS, see Koczot et al., 2004; Markstrom et al., 2016) to address three research questions: First, what shifts do droughts cause in the precipitation-runoff relationship of rain-snow catchments in a Mediterranean climate? Second, do these shifts affect water-balance predictive skill in basins with different predominant geology? Third, what is the catchment mechanism causing shifts during droughts as opposed to wet periods?

## 10  2   Methods

We focused on the Feather River upstream of Oroville Lake in the Sierra Nevada of California ($\sim$9300 km$^2$, see Figure 1) and on three of its sub-basins with contrasting geology (see Section 2.1 for details). Water from the Feather is both exploited locally for hydropower production by Pacific Gas & Electric (PG&E, see Freeman, 2011) and impounded by Oroville Dam to support water supply across the state through the State Water Project (Huang et al., 2012).

Our research followed three main steps (Sections 2.3.1 to 2.3.3). First, we quantified shifts during droughts in the observed precipitation-runoff relationship of the three (sub-)basins with serially complete full-natural-flow data (see Section 2.2 for details about full-natural-flow data). Second, we assessed the performance of the PRMS hydrologic model in predicting full-natural flow in all (sub-)basins and in particular during droughts in order to gain insight into the potential impact of these periods on predictive accuracy. Third, we identified the driver of predictive accuracy during droughts and its potential relation-

ship with shifts in precipitation-runoff relationship by comparing observed and simulated basin-wide mass-balance indexes ($P, ET, \Delta S, Q$) in the (sub-)basins with serially complete full-natural-flow data.

We focused on water years 1970 to 2015 due to both data availability and data quality. The water year is defined as October 1st to September 30th and it is indicated with the calendar year in which it ends (e.g., water year 2015 went from October 1st, 2014 to September 30, 2015). We defined drought water years following the official declarations of the State of California:

1976-77, 1987-92, 2007-09, and 2012-15 (see https://water.ca.gov/Water-Basics/Drought, visited July 19, 2019).

### 2.1   Study area

The climate of the Feather is Mediterranean, with dry summers and wet winters. Elevation ranges from $\sim$250 m above sea level (ASL) at Oroville Dam (the outlet of the basin) to $\sim$2900 m ASL at Mt. Lassen, but most of the catchment lies below 2000 m ASL (Koczot et al., 2004). Mixed rain-snow and rain-on-snow events are frequent across the basin (Koczot et al., 2004).

The water balance of the Feather has experienced recent warming-related changes, including declining runoff and peak snow accumulation (Freeman, 2011, 2012), forest growth (Freeman, 2011), and a rise in the rain-snow transition line (Hatchett et al., 2017).





The Feather is the most northern of the thirteen basins draining from the Sierra Nevada into the Sacramento-San Joaquin valley (see Harrison and Bales, 2016). Contrary to most of these catchments (Freeman, 2012; Harrison and Bales, 2016), some headwaters of the Feather lie in the eastern, rain-shadowed side of the Sierra divide (Figure 1): mean annual precipitation thus ranges from ∼2800 mm in the western side of the basin to less than 800 mm in the eastern side. Because low precipitation has been suggested as a key predictor of shifts during droughts (Saft et al., 2015), this basin is an ideal case study to answer our questions.

Our study considered two spatial scales (Figure 1): the Feather at Oroville Dam and three of its headwater sub-basins: Almanor (∼1100 km$^2$, 1400-2900 m ASL, rain-shadowed), East Branch (∼2600 km$^2$, 725-2550 m ASL, rain-shadowed), and Middle Fork (∼2700 km$^2$, 480-2660 m ASL, partially rain-shadowed). Hydrologic studies on the Feather River at Oroville are abundant (see for example Tang and Lettenmaier, 2010; Rosenberg et al., 2011; Huang et al., 2012; Anghileri et al., 2016, and references therein), whereas headwater sub-basins have rarely been studied as stand-alone catchments (see examples in Freeman, 2011; Wayand et al., 2015; Sun et al., 2019).

The Almanor sub-basin lies at the intersection between the granitic Sierra Nevada and the volcanic Cascade and is thus dominated by a porous, volcanic bedrock (see Figure 1). This sub-basin is mostly fed by subsurface flow (Freeman, 2008) and has exhibited a 30% decline in water-year inflow to Almanor Lake (located at the outlet of this sub-basin) since the 1960s. This drop is attributed to missed groundwater-recharge opportunities due to decreasing snow accumulation (Freeman, 2010). The geology of the East Branch and the Middle Fork includes impervious granitic outcrops similar to the rest of the Sierra Nevada. Water supply in these two sub-basins is dominated by surface runoff (Freeman, 2008), but the East Branch is significantly drier than the Middle Fork because it is fully rain shadowed.

## 2.2 Data

Data include daily full-natural flow at the outlet of the four (sub-)basins under study; in-situ precipitation, air temperature, and snow water equivalent ($SWE$) at various temporal resolutions; and annual spatially distributed water-balance indices of precipitation, evapotranspiration, and variation in sub-surface storage.

Full-natural (or unimpaired) flow is a mass-balance reconstruction of water supply as if no dam or other man-made infrastructure affected it (He et al., 2017). For the Almanor and East-Branch sub-basins, these data were provided by Pacific Gas & Electric (PG&E) at daily resolution for water years 1970 through 2017. For the Middle Fork sub-basin and for the Feather River at Oroville, data were obtained from the California Department of Water Resources (DWR) at daily and monthly resolution, respectively (water years 1987 to 2018 and 1985 to 2018, respectively).

In-situ daily precipitation from ten stations and daily maximum and minimum air temperature from three stations across the Feather river basin were obtained from PG&E, which routinely uses them as input for the PRMS hydrologic-forecasting model (see details in Koczot et al., 2004). These data were quality checked and serially gap-filled by the company (water years 1970 to 2017). Additional data of monthly in-situ precipitation and manual snow water equivalent were downloaded from the California Data Exchange Center (https://cdec.water.ca.gov/, visited July 19, 2019) to study drought characteristics across the Feather (see Table S1-S2 in the Supporting Information).





Spatially distributed annual precipitation ($P$) was from the Parameter-elevation Relationships on Independent Slopes Model (PRISM, see Daly et al., 2008). Spatially distributed $ET$ was estimated using a generalized additive model (GAM) between single-term power-function transformations of Landsat-based annual-averaged NDVI (Normalized Difference Vegetation Index, 30 m) and the average of the current and previous year's precipitation (Rungee et al., sub). The single-term power trans-
formations were developed by individually regressing the NDVI and PRISM data with flux-tower measurements of evapotranspiration (Rungee et al., sub). Variation in basin-wide subsurface storage was estimated as the residuals of $P - ET - Q$, where $P$ is basin-wide mean-annual PRISM-based precipitation, $ET$ is basin-wide mean-annual GAM-estimated evapotranspiration, and $Q$ is annual full-natural flow. Landsat-based data were available for water years 1985-2016; PRISM maps were processed for the same period. PRISM data have a pixel size of 800 m, which we downscaled to 30 m using a nearest-neighbor approach
to match that of Landsat.

## 2.3   Analyses

### 2.3.1   Shift in precipitation - runoff relationship

We detected shifts in the precipitation-runoff relationship by fitting a multivariate regression across annual cumulative full-natural flow (target variable), basin-wide annual precipitation, and a categorical variable denoting drought and non-drought
15   years (Saft et al., 2016a; Tian et al., 2018):

$$Q_{\mathrm{BC}} = b_0 + b_1 I + b_2 P + \epsilon, \tag{1}$$

where $I$ is a categorical drought variable (1 for drought years and 0 for non-drought years), $b_0, b_1, b_2$ are regression coefficients, $\epsilon$ is noise, and $Q_{\mathrm{BC}}$ is annual full-natural flow transformed according to a Box-Cox transformation following the arguments in Saft et al. (2016a, b); Tian et al. (2018):

$$Q_{\mathrm{BC}} = \frac{Q^\lambda - 1}{\lambda}. \tag{2}$$

While $\lambda$ should in principle be estimated from data to ensure linearity and heteroscedasticity (Sarkar, 1985), we assumed $\lambda = 0.25$ as suggested by Santos et al. (2018) and references therein.

If different from zero, parameter $b_1$ indicated a shift of the precipitation-runoff relationship during droughts. This parameter is usually negative, as shifts during droughts tend to decrease runoff compared to precipitation deficit alone (Saft et al., 2016a,
b). We assessed the statistical significance of coefficient $b_1$ and concluded that the shift during droughts was statistically significant if the sign of the confidence bounds agreed (Kottegoda and Rosso, 2008). We performed this analysis for the Feather River at Oroville (1985-2015) and the Almanor and East-Branch sub-basins (1970-2015), for which we had serially complete time-series of annual full-natural flow.

Rather than directly using PRISM maps to estimate basin-wide precipitation, we tilted their monthly mean surfaces using
precipitation data at the ten in-situ stations available to this study (see again Section 2.2). This operational procedure (called





DRAPER) is routinely used by PG&E forecasters on the river to force PRMS and is believed to provide more representative precipitation patterns for this basin than simply using PRISM surfaces (see Koczot et al., 2004; Donovan and Koczot, 2019, for details about the DRAPER algorithm).

We estimated the relative magnitude of the shift in precipitation vs. runoff ($M_Q$) for each (sub-)basin with serially complete

time-series of annual full-natural flow by using the approach suggested by Saft et al. (2016b):

$$M_Q = \frac{Q_{\mathrm{dry,P_I}} - Q_{\mathrm{dry,P}}}{Q_{\mathrm{dry,P}}}, \tag{3}$$

where $Q_{\mathrm{dry,P_I}}$ is the (predicted annual) full-natural flow for a representative (annual) precipitation during dry periods according to the shifted precipitation-runoff relationship (Equation 1, $I = 1$), while $Q_{\mathrm{dry,P}}$ is the full-natural flow for the same precipitation according to the non-shifted relationship (Equation 1, $I = 0$). We assumed as representative annual precipitation

the mean between average and minimum annual precipitation across the entire period of record (see more details, including a schematic, in Saft et al., 2016b). Here again, we used DRAPER to estimate this representative precipitation, while full-natural flow in Equation 3 was not transformed.

Saft et al. (2016b) originally proposed $M_Q$ to quantify the impact of the decade-long Millennium drought in south-eastern Australia ($\sim$1997-2009). The four Californian droughts under study were significantly shorter, so we applied Equation 3 by

aggregating all drought years in one sample. We also quantified shifts for single droughts ($m_Q$) by assuming $Q_{\mathrm{dry,P_I}}$ to be the observed, average annual full-natural flow across each drought, and $Q_{\mathrm{dry,P}}$ to be the expected annual full-natural flow according to the non-shifted precipitation-runoff relationship (Equation 1, $I = 0$) and a reference annual precipitation equal to the average across each drought.

### 2.3.2   PRMS performance during droughts: flow

PRMS is a semi-distributed hydrologic model that solves mass and energy conservation across a given basin by discretizing it into Hydrologic Response Units (HRUs), regions of the basin that are assumed homogeneous (Markstrom et al., 2015). The model requires, as a minimum, inputs of daily precipitation and maximum-minimum temperature at one location, from which these data can be distributed to the grid of HRU centroids (Markstrom et al., 2015). In the Feather River PRMS model, air temperature from three stations are distributed using monthly lapse rates. Precipitation is distributed using the DRAPER method

as outlined in Section 2.3.1 (Koczot et al., 2004; Donovan and Koczot, 2019). Processes simulated include precipitation-phase partitioning, precipitation interception and storage by canopy, evapotranspiration, radiation distribution, snow accumulation and melt, infiltration and surface runoff, interflow, groundwater storage, and baseflow.

PRMS was calibrated and evaluated over the Feather River in the early 2000s by mostly focusing on full-natural-flow data between 1971 and 1997 (see Koczot et al., 2004, for more details, including specific modules used by the model). While PRMS

has been updated since then (the current version is 5 – June 2019), the model is currently set up for this river in version 2. The main differences between more recent versions and version 2 are the sub-surface components: version 2 separates sub-surface storage into superficial soil (including the recharge zone), a deeper sub-surface reservoir, and groundwater (Koczot et al.,



2004); more recent versions of PRMS consider a process-based separation into capillary, preferential, and gravity storage in addition to groundwater (Markstrom et al., 2015). For this study, the representation of sub-surface processes in PRMS version 2 was assumed to be sufficiently representative of many conceptual models: for example, this version was implemented in inter-comparison tools like the Framework for Understanding Modeling Errors (FUSE, see Clark et al., 2008).

PRMS performance for full-natural flow was quantified using three different metrics: water-year Nash-Sutcliffe Efficiency (NSE), annual relative bias (relative to observations), and observed vs. simulated climate elasticity of streamflow. Because full-natural flow is prone to large errors, we smoothed the data and simulations using a five-day window before computing performance metrics.

    NSE benchmarks the squared errors of simulations of a target variable (in our case, daily full-natural flow for each water year)
against those obtained by using a long-term mean (Nash and Sutcliffe, 1970). The choice of this "long-term mean" can yield very sensitive results (Schaefli et al., 2007). In the Feather River basin, full-natural flow shows a large inter- and intra-annual variability (see some examples in Koczot et al., 2004), implying that a mean across all water years would be a particularly poor benchmark (resulting in overoptimistic NSE values). On the other hand, a water-year mean would be an excellent predictor during dry years and a very poor predictor during wet years. In order to limit these spurious results, we benchmarked PRMS
using a mean across all years from the same type according to the classification used by PG&E (henceforth, $NSE_{yrt}$; see Georgakakos et al., 2012, for some context on year types in California). We also computed the Nash-Sutcliffe Efficiency using log-transformed values ($LogNSE_{yrt}$) as this metric is more sensitive to low flow (Santos et al., 2018).

    Climate elasticity measures the sensitivity of annual streamflow (in this case, full-natural flow) to changes in a relevant climate variable, usually precipitation and potential evapotranspiration (Andréassian et al., 2016; Cooper et al., 2018). Compared
to NSE and bias, contrasting observed and simulated elasticity quantifies the performance of a hydrologic model in simulating inter-annual variability in streamflow and its relation to external forcings. Here, we considered absolute elasticity (nondimensional): $e_{Q/P}$ and $e_{Q/PET}$ are absolute elasticity to precipitation and potential evapotranspiration, respectively. Elasticity for both simulated and observed full-natural flow was computed in a bivariate linear framework using the approach proposed by Andréassian et al. (2016). As an independent benchmark, we also computed theoretical elasticity based on the Turc-Mezentsez
formula (see again Andréassian et al., 2016). Similarly to shifts in precipitation-runoff relationships (see Section 2.3.1), we computed elasticity for the (sub-)basins with serially complete time-series of annual full-natural flow, that is, the Feather River at Oroville (1985-2015) and the Almanor and East-Branch sub-basins (1970-2015). We again used DRAPER to estimate basin-wide precipitation; potential evapotranspiration was estimated using the Jensen-Haise approach in PRMS (Koczot et al., 2004).

### 2.3.3   PRMS performance during droughts: water balance

We quantified the performance of PRMS for the four components of the annual water balance by adapting Equation 1 to fit a regression between observed and simulated water-balance components during drought and non-drought years (period 1985-2015, see Section 2.2 for data availability of Landsat data products):



$$P_{\text{obs}} = b_{0,P} + b_{1,P}I + b_{2,P}P_{\text{sim}} + \epsilon_P \tag{4a}$$

$$ET_{\text{obs}} = b_{0,ET} + b_{1,ET}I + b_{2,ET}ET_{\text{sim}} + \epsilon_{ET} \tag{4b}$$

$$\Delta S_{\text{obs}} = b_{0,\Delta S} + b_{1,\Delta S}I + b_{2,\Delta S}\Delta S_{\text{sim}} + \epsilon_{\Delta S} \tag{4c}$$

$$Q_{\text{obs}} = b_{0,Q} + b_{1,Q}I + b_{2,Q}Q_{\text{sim}} + \epsilon_Q, \tag{4d}$$

where, for example, $P_{\text{obs}}$ and $P_{\text{sim}}$ are observed and simulated basin-wide annual precipitation, while $b_{0,P}$, $b_{1,P}$, and $b_{2,P}$ are regression coefficients, respectively. If $b_{1,P}$ was statistically different from zero, then PRMS performance for precipitation during droughts was statistically different from that during non-drought years. Similar definitions apply to the other water-balance components in Equation 4. This analysis was carried out for the (sub-)basins with serially complete time-series of annual full-natural flow (see Section 2.3.1).

Observed precipitation was assumed equal to the original PRISM maps in absence of other independent, distributed sources. Observed $ET$ was derived from the GAM-estimated maps introduced in Section 2.2, while observed $\Delta$S was estimated closing the water balance with observed full-natural flow (see again Section 2.2).

In addition to the standard four balance terms in Equation 4, PRMS includes a groundwater sink. This term is used to account for (often unknown) water losses in the sub-surface portion of the water balance. Because this sink together with $ET$ represents the only internal water loss in the model, it was summed to $ET_{\text{sim}}$ to fit a regression with observed evapotranpiration (note that this groundwater sink was set to zero in the original calibration of the Almanor sub-basin).

## 3 Results

### 3.1 Drought hydro-climatic summary

Average minimum daily air temperature at the three index stations of Canyon Dam (1390 m ASL), Quincy (1066 m ASL), and Bucks Creek Powerhouse (536 m ASL) showed a statistically significant increasing trend between water years 1970 and 2015 (Kendall $\tau = 0.5$, p-value $< 0.01$, $\alpha = 0.05$, Sen's slope $= 0.0448 \pm 0.0134$°C yr$^{-1}$, that is, $\sim$2°C in 45 years, Figure 2). Neither average maximum daily air temperature nor annual precipitation presented a statistically significant trend (p-value $= 0.57$ and 0.99, respectively – statistics for precipitation refer to median values across all stations, see Figure 2). April-1st SWE also had no significant trend for $\alpha = 0.05$, but its p-value was significantly smaller (0.09, statistics again refer to median values across all stations, see Figure 2). The ratio between median April 1st SWE and median annual precipitation showed a statistically significant, yet slight, shift from snow to rain (Kendall $\tau = -0.22$, p-value $= 0.0279$, $\alpha = 0.05$, Sen's slope $= -0.0055$ yr$^{-1}$).

The four droughts under study had very different hydro-climatic characteristics (Figure 2 and Table 1). The 1976-77 drought was both the coolest and driest in our record ($\sim$56% of average annual precipitation compared to 2012-15), and as a result 1976 and 1977 were the fourth and first driest water years in the State's historical record at that time (DWR, 1978). Gauged flow





from the Feather at Oroville was 43% and 24% of (contemporary) average for 1976 and 1977, respectively, the latter being a new record. Storage of Oroville Lake on October 1, 1977 was ~37% of the norm (DWR, 1978).

At the other end of the spectrum, the 2012-15 drought was the warmest and (together with the short 2007-09 drought) the wettest during our study period (Figure 2 and Table 1). As a result, average April-1st SWE/P significantly declined during this

drought compared to the other three (0.26 as opposed to ~ 0.50). A condition of comparatively high precipitation but lacking snow storage has been recently defined a warm snow drought (Harpold et al., 2017). In between, the 1987-92 drought was the longest one (6 years), with five years classified as critically dry and one (1989) as dry (DWR, 1993). Average minimum temperature during this drought was ~0.89 °C higher than the 1976-77 drought.

While all droughts decreased annual water supply (see Sections 3.2), runoff seasonality was generally preserved between

drought and non-drought years (Figure 3), with peak flow occurring during winter and spring due to precipitation and snowmelt, and low flow occurring during the dry summer-fall season. While the volcanic, subsurface-flow-fed Almanor sub-basin and the surface-runoff-dominated East-Branch sub-basin displayed comparable peaks in full-natural flow during winter, the latter had a significantly lower flow during summer than the former. This higher summer flow in the volcanic Almanor sub-basin compared to the granitic East Branch was consistent between drought and non-drought years.

## 3.2  Shift in precipitation vs. runoff

The four droughts under study caused a shift in the precipitation-runoff relationship for both the two headwater sub-basins with complete annual data (Almanor and the East Branch) and the Feather River at Oroville (Figure 4). This shift means that the decrease in runoff observed during droughts in these (sub-)basins was larger than what could be explained by precipitation deficit alone (see Section 2.3.1 for details about the definition of shift). The 95% confidence bounds for $b_1$ were -1.51 and

20  -0.34; -2.29 and -1.00; and -1.45 and -0.29, respectively (see Equation 1 for a definition of $b_1$), implying that this shift was statistically significant for all (sub-)basins. The magnitude of the shift (calculated using Equation 3) was -18%, -39%, and -18% compared to precipitation allocation to runoff during non-drought years, respectively. Runoff from the volcanic Almanor sub-basin was thus more resilient to shifts during droughts than that from the East Branch, even if shifts were significant in all the (sub-)basins investigated.

The magnitude of these shifts varied from drought to drought and across (sub-)basins (Table 2). In the volcanic Almanor sub-basin, the largest shift corresponded to the 1987-92 drought (-25%), the longest in our record. For the surface-runoff-dominated East Branch, the largest shift was caused by the 1976-77 drought (-51%), the shortest, driest, and coolest in our record (see again Table 1); in general, this sub-basin consistently showed the largest shift during all droughts across all (sub-)basins. For the Feather River at Oroville, the largest shift corresponded to the recent 2012-15 drought (-22%), the warmest in our record

(note that no data was available for this basin during the 1976-77 drought). The 2007-09 drought showed the smallest shift in all basins under study.





### 3.3 PRMS performance for flow

Both $NSE_{yrt}$ and $LogNSE_{yrt}$ significantly decreased during droughts (Figure 5). The difference between average $NSE_{yrt}$ and $LogNSE_{yrt}$ during drought vs. non-drought years and the four (sub-)basins was -0.6 and -0.4 (Almanor); -1 and -0.4 (East Branch); -0.9 and -0.2 (Middle Fork); and -0.4 and -0.1 (Feather at Oroville). The performance during isolated dry years was

5 better than during droughts (e.g., see water years 1994 or 2001 in Figure 5). This decline in PRMS performance was comparable between droughts that were part of the calibration period (1970-1997) and those that occurred after 1997.

For the 1976-77 drought, $LogNSE_{yrt} < NSE_{yrt}$ in both the volcanic Almanor sub-basin and the East Branch. $NSE_{yrt}$ is very sensitive to high flows, while $LogNSE_{yrt}$ puts more weight on low flows (Santos et al., 2018): low flows were thus the driver of performance drops during the 1976-77 drought. In the East Branch, $NSE_{yrt}$ during that drought was even comparable

to non-drought years. For the 1987-92 and the 2012-15 droughts, $NSE_{yrt} < LogNSE_{yrt}$ in both the Almanor sub-basin and the East Branch: high-flow peaks such as those during winter precipitation events or spring snowmelt were thus the main performance driver during these wetter droughts. The performance during the 2007-09 drought was only slightly below non-drought-year standards. The response time of $NSE_{yrt}$ to droughts in the Almanor sub-basin was somewhat slower than in the other (sub-)basins, a good example being the decadal drop during the 1980s - early 1990s.

Observed annual full-natural flow was generally overestimated during all droughts but the 1976-77 one, for which PRMS severely underestimated water supply for both the Almanor and the East Branch sub-basins (relative biases of -0.44 and -0.86, respectively, Figure 6). Inter-annual variability in relative bias was larger in the surface-runoff-dominated East Branch and Middle Fork than in the volcanic Almanor sub-basin. The 2007-09 drought returned relative biases in the East Branch and Middle Fork that were in line with non-drought years.

PRMS overestimated the absolute value of climate elasticity of streamflow to both annual precipitation and potential evapotranspiration (Table 3 and Figure S1 in the Supporting Information). In particular, both observations and simulations showed a statistically significant positive elasticity to precipitation, but observations were closer to the theoretical elasticity according to the Turc-Mezentsev formula. With regard to annual potential evapotranspiration, observations did not show any statistically significant elasticity, whereas PRMS-based elasticity was statistically significant for both the Almanor and the East Branch

sub-basins. The largest overestimation of elasticity corresponded to the volcanic Almanor sub-basin. We interpret the fact that modeled $e_{Q/PET}$ was unexpectedly positive as likely spurious and related to the large scatter and weaker correlations between potential evapotranspiration and modeled full-natural flow compared to precipitation vs. modeled full-natural flow (Figure S1, univariate correlation coefficient for precipitation and potential evapotranspiration vs. modeled full-natural flow being $\sim 0.95$ and $\sim 0.43$, respectively).

### 3.4 The observed vs. modeled water balance

PRMS-modeled and PRISM-based basin-wide precipitation were visually in very good agreement, both in terms of annual values and in terms of inter-annual variability (Figure 7 and S2 - S3 in the Supporting Information). This was expected, as PRMS uses PRISM as a starting point to distribute precipitation across the watershed (DRAPER method, see Section 2.3.2),





which does not significantly affect annual precipitation totals. On the other hand, the model significantly underestimated annual estimated evapotranspiration, even if this underestimation was partially compensated for by the groundwater sink (Figure 7 and S2 - S3). Also, the model systematically underestimated both the absolute value and the interannual variability of changes in sub-surface storage, in particular for the Almanor sub-basin (Figure S3) and for the Feather River at Oroville (Figure 7); PRMS

failed to reproduce the multi-decadal decline in storage observed in all (sub-)basins as a result. While the observed changes in sub-surface storage used in this paper to evaluate PRMS may suffer from unquantifiable uncertainty across precipitation, full-natural flow, and GAM-estimated evapotranspiration, this decline was confirmed by other soft data collected on the river (Freeman, 2011) and agrees with a general trend of declining summer low flows in the Maritime Western U.S. (Cooper et al., 2018).

Among the three water-balance components determining full-natural flow, $ET$ was the only one for which the performance of PRMS during drought and during non-drought years were statistically different in all (sub-)basins (see Figure 8 and Table 4). Conversely, the performance for precipitation and for changes in sub-surface storage were statistically different in only one sub-basin each: the East Branch for $P$ and Almanor for $\Delta$S, respectively. As expected (see Section 3.3), differences in PRMS performance for full-natural flow during drought vs. non-drought years were statistically significant in all basins. Thus, $ET$

was the only water-balance component (besided $Q$) that was systematically misrepresented during droughts.

The statistically different performance of PRMS for $ET$ during droughts was confirmed even when comparing simulated and observed average $ET$ (including groundwater sink) over temporal windows of two, three, and four years (see Figure S4 in the Supporting Information), which may be more appropriate time scales for this evaluation because of the possible temporal lag between vegetation greenness and $ET$ (Goulden et al., 2012). Results in Figure 8 and Table 4 were also confirmed when

only considering $ET$ rather than the sum of $ET$ and the groundwater sink (see Figure S5 in the Supporting Information and Section 2.3.3 for details about the groundwater sink).

## 4 Discussion

Previous studies about drought-driven shifts in precipitation vs. runoff have mostly focused on rainfall-dominated regions like Australia (see Saft et al., 2015, and references therein) or China (Tian et al., 2018), but we showed here that such shifts

may also occur in mixed rain-snow catchments in a Mediterranean climate, regardless of predominant geology (volcanic, subsurface-flow-dominated or transitional-to-granitic, surface-runoff-dominated). In agreement with previous findings by Saft et al. (2016a), we also found that only droughts corresponding to significant shifts in precipitation-runoff relationships translated into poorer performances for a semi-distributed hydrologic model (PRMS), meaning that understanding this drop in accuracy may shed light on the mechanism behind the observed shifts. Because $ET$ is the only water-balance component

(besides $Q$) yielding statistically different performances during droughts vs. non-drought years, $ET$-drought feedback mechanisms are the most likely driver of shifts in water supply in a Mediterranean, mixed rain-snow climate, and failure to fully account for these mechanisms results in the predictive inaccuracy of the model.





Shifts in precipitation vs. runoff have attracted increasing interest since at least the recent Millennium drought in south-eastern Australia, where Saft et al. (2016b) estimated maximum shifts in precipitation vs. runoff in the order of 80-100%. If and where these shifts occur, runoff will decrease more than what would be predicted based on a precipitation-runoff relationship trained using non-drought years (Saft et al., 2016b). With hydrologic models that are often biased toward better reproducing

high flows (Staudinger et al., 2011) and climate-change scenarios that predict increasing aridity in several regions of the world (Cayan et al., 2010), understanding the mechanisms behind these shifts and the adequacy (Gupta et al., 2012) of models in such conditions has profound implications for water resources management and water security.

In this Section, we elaborate on three of these implications: first, why is $ET$ consistently misrepresented during droughts? Second, are shifts in precipitation vs. runoff common across all basins of the Sierra Nevada? Third, what are the lessons learned

to improve water-supply simulations in drought-prone regions?

### 4.1   Why $ET$ is misrepresented during droughts: climate elasticity of evapotranspiration

Annual errors for basin-wide $ET$ were due to (1) a systematic bias (∼160 mm less simulated $ET$ than observed) and (2) an annually variable component (Figure 9, results refer to the Feather at Oroville). The systematic bias could be explained by an underestimation of plant-accessible water storage, a recurring source of uncertainty in the Sierra Nevada (Klos et al., 2018).

Like any hydrologic model that is calibrated on streamflow, however, the annual performance of PRMS for $Q$ is relatively insensitive to systematic biases in internal fluxes like $ET$ as these can be offset by other fluxes during calibration (a good example being the groundwater sink, see Figure 7). This means that the drop in modeling accuracy during droughts and its relationship with shifts in precipitation-runoff relationships is related to the annually variable component. This component of the error was indeed qualitatively correlated with drought vs. non-drought years: PRMS tended to relatively overestimate

$ET$ during droughts and to underestimate it during non-drought years (see again Figure 9). We did not find any qualitative correlation between errors for $\Delta S$ and drought years (Figure S6), implying again that $ET$ is the driver of predictive inaccuracy for this model during droughts.

Bales et al. (2018) suggested three sources of conceptual uncertainty with regard to how models simulate the drought water balance. The first is the already mentioned oversimplification of regolith storage and rooting depth and thus misrepresentation

of plant-accessible water storage (see also Rungee et al., 2018). The second is a lack of proper parametrization of the feedback between evaporative demand and stomatal closure. The third is the representation of vegetation as a static layer with no dynamic response to stresses. From this perspective, Figure 10 shows that relative differences between observed and simulated $ET$ during droughts across the Feather at Oroville (relative to the systematic bias) were somewhat correlated with maximum and minimum annual temperature ($r$ = -0.45 and -0.57, respectively) and with observed relative changes in storage ($r = 0.23$

– also relative to the corresponding systematic bias, see Figure S6). The correlation with annual precipitation during droughts was much smaller ($r = 0.1$). Correlations during non-drought years were visually similar to those during droughts (Figure 10), and differences compared to drought years should be interpreted with caution given the small number of available data points.

While none of these correlations is strong enough alone to explain modeling errors, they collectively point to interactions across atmospheric demand for moisture, $ET$, and sub-surface storage as the source of conceptual uncertainty behind the





misrepresentation of $ET$ during droughts. The overall picture is that PRMS relatively overestimates $ET$ during years characterized by comparatively cold conditions and a relative replenishment of sub-surface storage (both conditions that would in fact decrease $ET$), and underestimates $ET$ during comparatively warm years characterized by a relative drawdown of storage (both conditions that would in fact increase $ET$). In other words, the model underestimates the multi-year response of $ET$ to

climate *variability*, a property that we hereby define as *climate elasticity of evapotranspiration* and that emerges as a driver of water supply in a Mediterranean climate (Bales et al., 2018). While tree mortality may also be a good explanatory variable for errors in $ET$, available data on this river only date back to the early 2000s and were thus too short to compute correlations.

## 4.2   Are shifts in precipitation vs. runoff common across the Sierra Nevada?

Our results show that shifts in precipitation-runoff relationship may take place both in volcanic, subsurface-flow-dominated and

in transitional-to-granitic, surface-runoff-dominated basins. This may seem counterintuitive as the volcanic Almanor sub-basin has a relatively small inter-annual variability in low flows that agrees with other studies in similar contexts (Jefferson et al., 2008; Tague et al., 2008; Cooper et al., 2018), and this variability is an important driver of shifts in Australian basins (Saft et al., 2016b). However, while the surface-runoff-dominated East Branch does return larger shifts both overall ($M_Q$) and for individual droughts ($m_Q$, see Section 3.2), both this sub-basin and the Almanor are rain-shadowed (aridity index of ~1.5 and

1.1, respectively). According to Saft et al. (2016b), pre-drought aridity is the most important predictor of shifts. This highlights that a higher summer flow does not necessarily provide resiliency to shifts in precipitation vs. runoff during droughts. That said, the volcanic sub-basin was still more sensitive to longer droughts than the surface-runoff-dominated sub-basin. This behavior may be related to the comparatively long time needed by groundwater-flow-dominated, slow-draining catchments to respond to a superficial precipitation deficit, which has also been shown to decrease elasticity of summer low flows to superficial inputs

(Cooper et al., 2018)

The magnitude of shifts in our case study were comparable to previous findings: Tian et al. (2018) reported an average shift of -24% in 18 rivers in China; Saft et al. (2016b) found a mode between -40% and -20% in Australia. By upscaling the analysis in the present study to the twelve other major rivers draining the western side of the Sierra Nevada to the California Central Valley, we found that eight of these twelve basins showed statistically significant shifts, on the order of -19% to -12%

(Figure 11 and Table S3, water years 1985-2018, data from PRISM and https://cdec.water.ca.gov/, visited July 19, 2019). The basins that did not show a statistically significant shift were the relatively small, low-elevation Cosumnes and Tule basins plus two other southern basins, the Kaweah and the Kern (see a map and a summary of hydrologic characteristics of each basin in Harrison and Bales, 2016). While the Kaweah and the Kern have high-elevation, snow-dominated headwaters, they also have significant rain-modulated low-elevation areas. Likewise, the low-elevation Cosumnes and Tule are mostly rain dominated. This

suggests that mixed rain-snow, Mediterranean basins in which rain plays a more prominent role in the annual water budget are less prone to shifts in the precipitation-runoff relationship. We interpret this as being because, in the snow-dominated basins where a significant shift is observed, snow-melt replenishes sub-surface storage later into the dry season and thus limits the dependence of evapotranspiration from deep sub-surface storage (Rungee et al., 2018), a key mechanism that also reduces low-flow elasticity to climate variability (Cooper et al., 2018) but that is greatly reduced during droughts.



The significance of these shifts was, however, sensitive to a number of methodological choices that are worth discussing given the relatively small amount of literature on statistically quantifying shifts in precipitation-runoff relationships. First, if the period 2016-2018 had been removed from computations in Figure 11 as was done for the Feather in Figure 4, only the Feather, the Tuolomne, and the San Joaquin would have returned a statistically significant shift. The magnitude of the shift was

more robust, with average differences between the estimates with or without the period 2016-18 on the order of 3%. Second, annual precipitation in Figure 11 was directly estimated from PRISM surfaces rather than by tilting them with index in-situ stations as done on the Feather (see Section 3.3). This choice was made for consistency reasons due to the lack of a comparable tool to DRAPER in the other basins of the Sierra. While results for the Feather River at Oroville using both approaches were visually in good agreement (not shown), the absolute magnitude of the shift for this river is smaller with original (-12%) than

with tilted PRISM data (-18%). It is challenging to assess which of the two indexes is closer to actual precipitation, but this comparison helps quantify the contribution of $P$ on the overall uncertainty in $M_Q$. Third, results were sensitive to the choice of using the Box-Cox transformation (Equation 2): by focusing on the Feather river (sub-)basins, only the East Branch and the Feather at Oroville would have returned a statistically significant shift using non-transformed full-natural flow data.

Overall, this discussion demonstrated that precipitation vs. runoff shifts are a common feature of mixed rain-snow catchments

in the Sierra Nevada of California. At the same time, results underscored the importance of explicitly including an uncertainty-sensitivity analysis when quantifying these shifts, especially because they are the result of differences across large numbers ($P$ and $Q$) that are particularly uncertain in mountain contexts (Avanzi et al., 2014).

### 4.3 How to achieve more robust runoff predictions during droughts?

While we considered only one model, PRMS, the conceptual-uncertainty source outlined in Figure 10 is a common feature

among hydrologic models that traces back to fundamental knowledge gaps in Critical Zone science such as the actual depth to which roots can access water in regolith (Klos et al., 2018). While it has been hypothesized that more observations could improve the performance of models during droughts (Chiew et al., 2014), the accuracy of PRMS for droughts during the calibration period was similar to that for droughts outside it. This version of PRMS was calibrated by mixing visual inspection and multiple objective functions such as root mean square error, bias, and relative error (Koczot et al., 2004), which may have

skewed model fitting toward high flows compared to multi-objective criteria like the Kling-Gupta Efficiency (Gupta et al., 2009) or low-flow-oriented metrics like LogNSE. More measurements of evapotranspiration in mountain regions could help better parametrize climate elasticity of evapotranspiration and thus support improved calibration by unraveling the role that this property plays in buffering the impact of precipitation deficit on runoff during droughts.

Shifts in the precipitation-runoff relationship of snow-dominated regions are particularly critical because in these contexts

snow plays a key role in both water supply and its seasonal predictability. In the western United States, water-supply forecasts are based on statistical regressions across full-natural flow, precipitation, and snowpack accumulation (Pagano et al., 2004; Rosenberg et al., 2011; Harrison and Bales, 2016). These forecasts play a key role in water-resources allocation across industrial and agricultural uses as well as freshwater supply (Harrison and Bales, 2016). While using runoff-to-date as a predictor and fitting different regression coefficients for different year types may partially correct for runoff deficit, these regressions do not





explicitly account for shifts in precipitation-runoff relationships during droughts (Harrison and Bales, 2016). Since shifts in precipitation-runoff relationships are common across the Sierra Nevada (Figure 11), we suggest embedding a shift predictor into these regressions as done in Equation 1 as a potential future step of this work.

The underestimation of runoff during the 1976-77 drought disagrees with the general consensus that these models tend to
overestimate water supply in regions where droughts shift the precipitation-runoff relationship (Saft et al., 2016a; Tian et al., 2018). Water year 1977 was still the driest among the 114 years on record for California in 2015 (DWR, 2015), and it was the last of three years of consecutively decreasing precipitation (Figure 2). Long dry periods may lead to a disconnection between soil and groundwater storage, which in turn may prevent recharge and favor direct surface runoff and interflow (see Saft et al., 2016b, and references therein). This condition of temporary hydrophobicity of soils and the subsequent slower-than-
expected recovery of soil-water storage (Sowerby et al., 2008) are not captured by PRMS. Neglecting this process may lead to erroneously allocating precipitation to the recharge zone (where it becomes available for evapotranspiration) or to groundwater; in either case, runoff would be underestimated. Here again, surface-to-subsurface mass and energy fluxes emerge as the most relevant knowledge gap in this field that would benefit from more targeted research.

## 5   Conclusions

Droughts cause a shift in the precipitation-runoff relationship of Mediterranean mixed rain-snow mountain basins of the Sierra Nevada of California. The magnitude of this shift is comparable to previous findings in rainfall-dominated semi-arid areas with year-around or summer-monsoon-dominated precipitation, which points to common feedbacks impacting the process across precipitation regimes. By comparing observed water-balance components during drought vs. non-drought years with those simulated by a hydrologic model, we identified some of these common feedbacks as being driven by the multi-year response
of evapotranspiration to climate and in particular to atmospheric demand for moisture (temperature) and to subsurface water storage. Surface-runoff-dominated catchments are prone to larger shifts in precipitation-runoff relationships than catchments dominated by subsurface flow because of the modulating effect of groundwater on the annual water balance of the latter. Snow-dominated basins are also more susceptible to shifts than rainfall-dominated basins because snow melt during the dry season limits evapotranspiration dependence on deep sub-surface storage – a mechanism that is greatly reduced during droughts. The
complex response of evapotranspiration to climate in mixed rain-snow Mediterranean catchments caused significant drops in performance for a semi-distributed hydrologic model (PRMS). Improved parametrizations of climate elasticity of evapotranspiration are thus highly needed to make models and water resources management more robust to droughts, especially in a warming and more variable climate.

*Competing interests.*  Authors have no competing interest.



*Code and data availability.* Sources of data used in this paper are reported in the main text (in particular in Section 2.2). The PRMS hydrologic model is available at https://www.usgs.gov/software/precipitation-runoff-modeling-system-prms.

*Acknowledgements.* This work was supported by the California Energy Commission under contract EPC-14-067, Pacific Gas & Electric Co, the California Department of Water Resources, and the UC Water Security and Sustainability Research Initiative. Tessa Maurer was
5  supported by the National Science Foundation Graduate Research Fellowship under Grant No. DGE 1106400. We would like to thank Kevin Richards (PG&E) for his continuous support and advice throughout this work.



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





**Table 1.** Average precipitation ($P$), Snow Water Equivalent ($SWE$), and maximum-minimum daily temperature ($T_{max}$ and $T_{min}$, respectively) across the four California droughts under study. Annual statistics are reported in Figure 2.

| Drought | $P$ (m) | $SWE$ (m) | $SWE/P$ (-) | $T_{max}$ (degC) | $T_{min}$ (degC)[a] |
|---|---|---|---|---|---|
| 1976-77 | 0.49 | 0.23 | 0.46 | 19.56 | 2.66 |
| 1987-92 | 0.79 | 0.39 | 0.51 | 20.08 | 3.55 |
| 2007-09 | 0.90 | 0.46 | 0.51 | 19.30 | 4.02 |
| 2012-15 | 0.84 | 0.22 | 0.26 | 20.04 | 4.58 |

[a] $P$ is average water-year precipitation during each drought across 13 stations on the Feather River (see Table S1 in the Supporting Information). $SWE$ is average April 1st SWE during each drought across 25 stations on the Feather River (see Table S2 in the Supporting Information). $T_{max}$ and $T_{min}$ are average annual maximum and minimum daily temperature during the drought at the three index stations used by the PRMS model for air-temperature input: Canyon Dam (1390 m), Quincy (1066 m), and Bucks Creek Powerhouse (536 m). Data sources: California Data Exchange Center (CDEC, https://cdec.water.ca.gov/, visited July 19, 2019) and Pacific Gas & Electric.





**Table 2.** Estimated shift in the precipitation-runoff relationship for single droughts on the Feather River (see Section 2.3.1).

| Drought | $m_Q$ Almanor (%) | $m_Q$ East Branch (%) | $m_Q$ Oroville (%) |
|---------|-------------------|-----------------------|---------------------|
| 1976-77 | -11 | -51 | – |
| 1987-92 | -25 | -36 | -21 |
| 2007-09 | -9 | -20 | -6 |
| 2012-15 | -18 | -47 | -22 |





**Table 3.** Modeled, observed, and theoretical climate elasticity of streamflow to annual precipitation ($e_{Q/P}$) and potential evapotranspiration ($e_{Q/PET}$) for the three (sub-)basins under study with complete annual full-natural flow data. Theoretical elasticity was computed according to the Turc-Mezentsev formula (Andréassian et al., 2016). The asterisk (*) denotes statistically significant elasticity values (that is, the sign of the confidence bounds agrees, 95% confidence level).

| (Sub-)basin | $e_{Q/P}$ (-) | | | $e_{Q/PET}$ (-) | | |
|---|---|---|---|---|---|---|
| | Modeled | Observed | Theoretical | Modeled | Observed | Theoretical |
| Almanor | 0.80* | 0.59* | 0.55 | 0.44* | -0.08 | -0.31 |
| East Branch | 0.68* | 0.56* | 0.33 | 0.21* | -0.08 | -0.14 |
| Oroville | 0.73* | 0.69* | 0.47 | 0.26 | -0.07 | -0.25 |





**Table 4.** Regression between observed and simulated water-balance components: confidence bounds of the regression parameter ruling shifts in performance during droughts (see Equation 4). $ET$ is the sum of $ET$ and the groundwater sink (see Section 2.3.3). The asterisk (*) denotes statistically significant elasticity values (that is, the sign of the confidence bounds agrees, 95% confidence level).

| (Sub-)basin | $b_{1,P}$ (mm) | $b_{1,ET}$ (mm) | $b_{1,\Delta S}$ (mm) | $b_{1,Q}$ (mm) |
|---|---|---|---|---|
| Almanor | -136 to +42 | -125 to -33* | +13 to +119* | -154 to -1* |
| East Branch | -187 to -14* | -81 to -17* | -57 to +41 | -129 to -17* |
| Oroville | -125 to +4 | -76 to -14* | -5 to +76 | -117 to -4* |



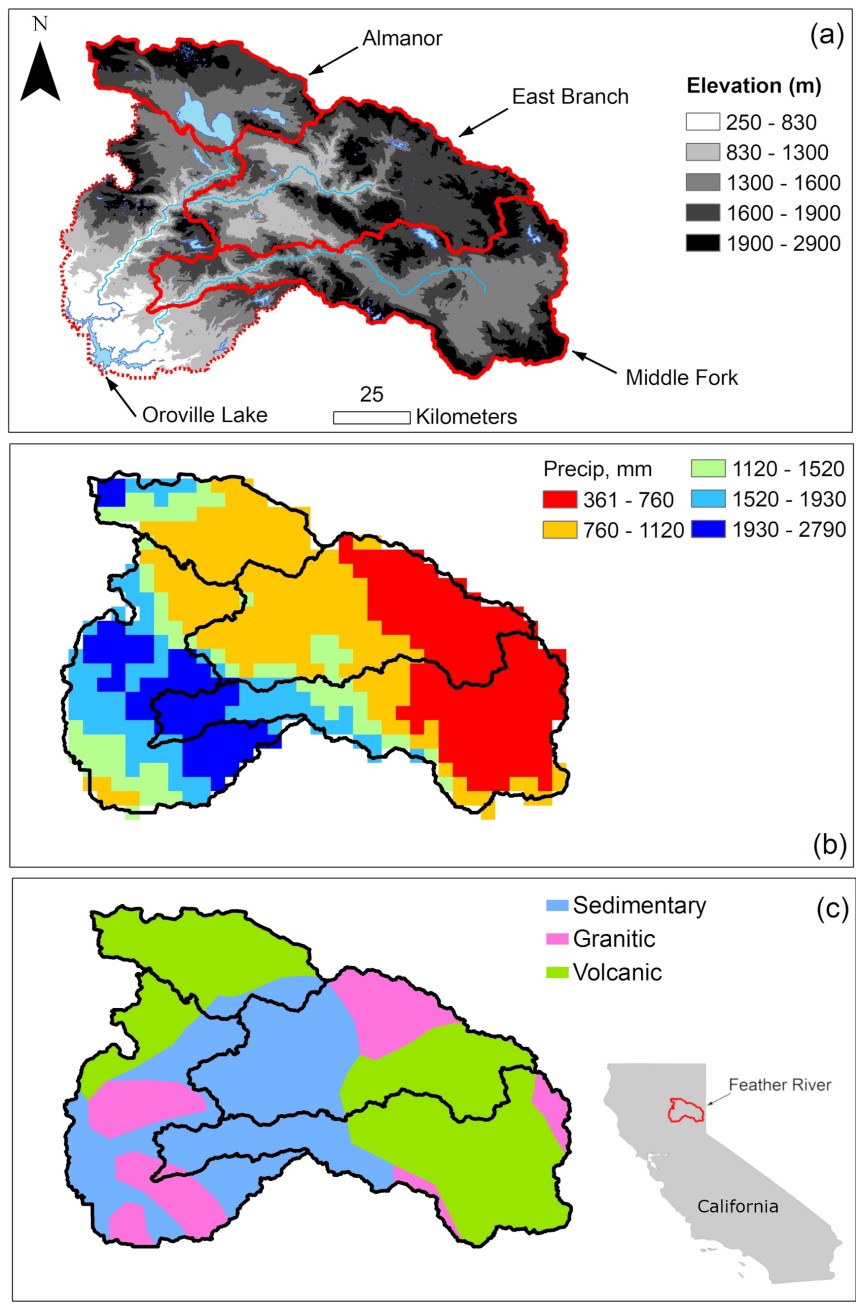

**Figure 1.** The Feather River at Oroville and its three headwater sub-basins under study (Almanor, East Branch, and Middle Fork): orography and hydrography (a), PRISM 1981-2010 average annual precipitation (b), predominant geology according to the USGS National Atlas (c).

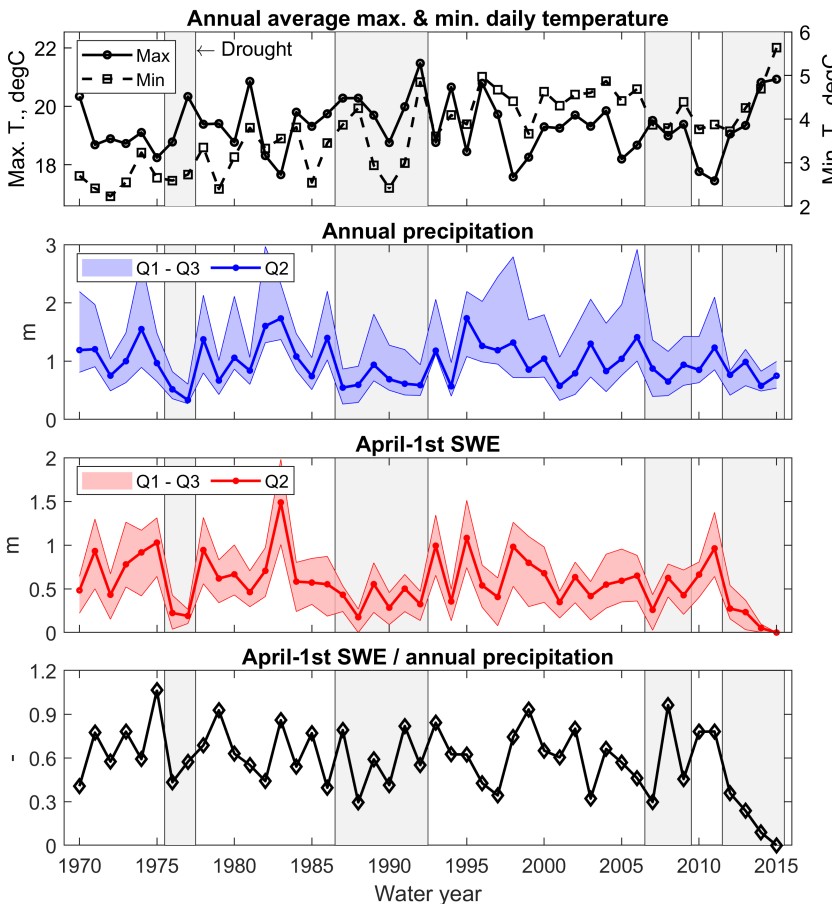

**Figure 2.** Hydroclimatic summary of the four most recent Californian droughts compared to non-drought years. Maximum and minimum daily temperature is an annual average of the three index stations used by the PRMS model for air-temperature inputs: Canyon Dam (1390 m), Quincy (1066 m), and Bucks Creek Powerhouse (536 m). Annual precipitation and April-1st SWE were computed using 13 and 25 stations across the Feather River, respectively (see Table S1 and S2 in the Supporting Information). The ratio between April-1st SWE and annual precipitation was computed with reference to spatial medians. Q2 is the spatial median, Q1 and Q3 are the two quartiles, respectively. Sources: California Data Exchange Center (CDEC, https://cdec.water.ca.gov/, visited July 19, 2019) and Pacific Gas & Electric.



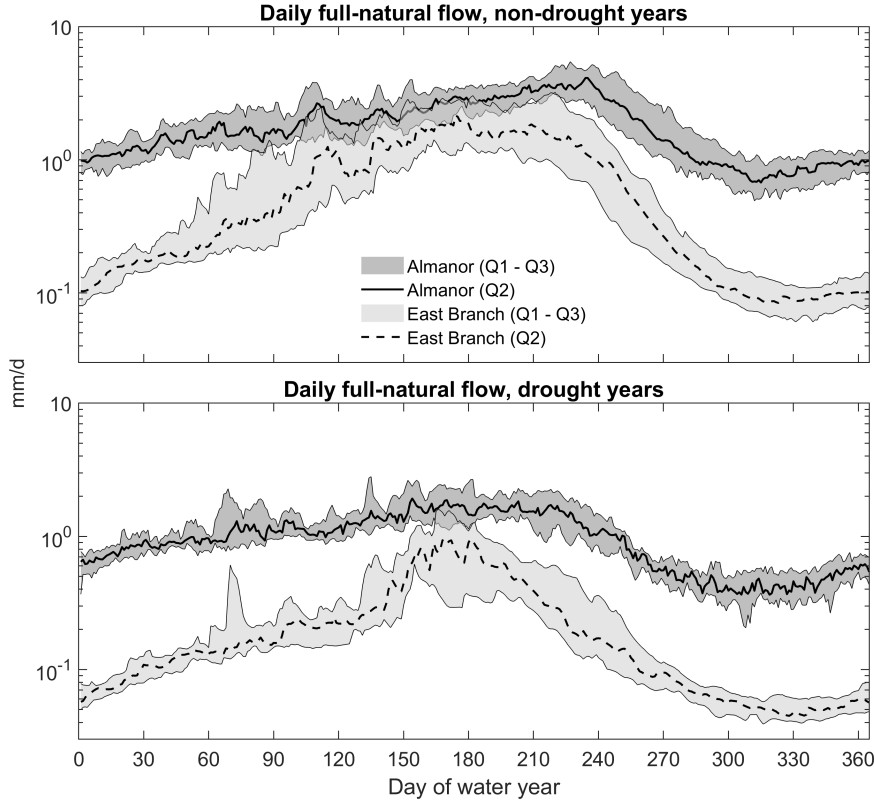

**Figure 3.** Daily median (Q2) and quartiles (Q1 and Q3) of observed full-natural flow during drought and non-drought years at the outlet of two headwater basins of the Feather River with contrasting geology. The Almanor subbasin is a predominantly volcanic, subsurface-flow-fed subbasin; the East Branch is transitional to granitic and surface-runoff-dominated. The y axis in in log-scale.




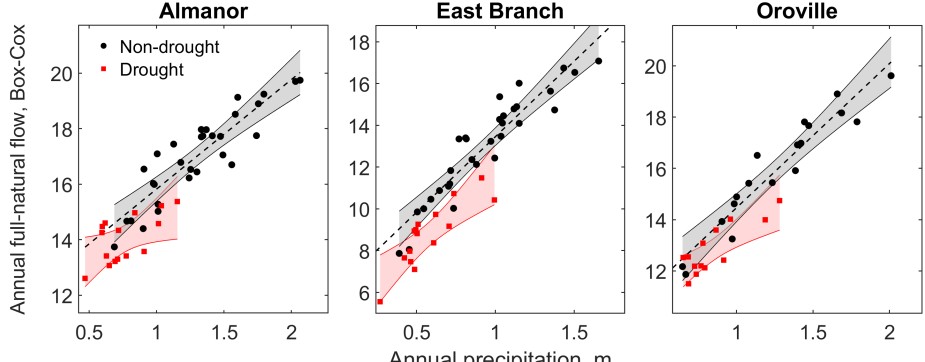

**Figure 4.** Precipitation-runoff relationship for drought (red) vs. non-drought (black) years and the three basins under study with complete annual runoff data (Almanor, East Branch, and Feather River at Oroville). The red and grey bands represent 95% confidence intervals for the regressions. The Box-Cox transformation for annual full-natural flow is introduced in Section 2.3.1, Equation 2.

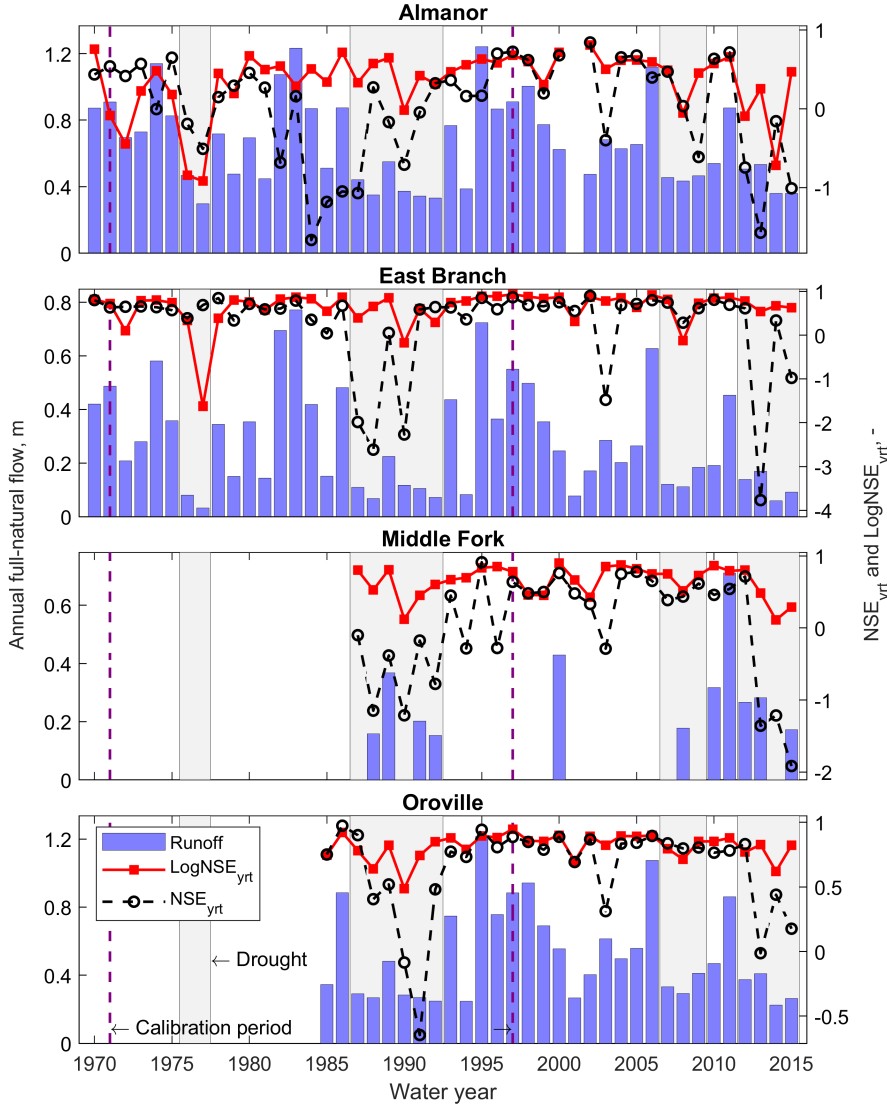

**Figure 5.** Water-year Nash-Sutcliffe Efficiency ($NSE_{yrt}$) and Log-Nash-Sutcliffe Efficiency ($LogNSE_{yrt}$) for PRMS-modeled daily full-natural flow. The blue bars represent observed annual full-natural flow at the outlet of each (sub-)basin.



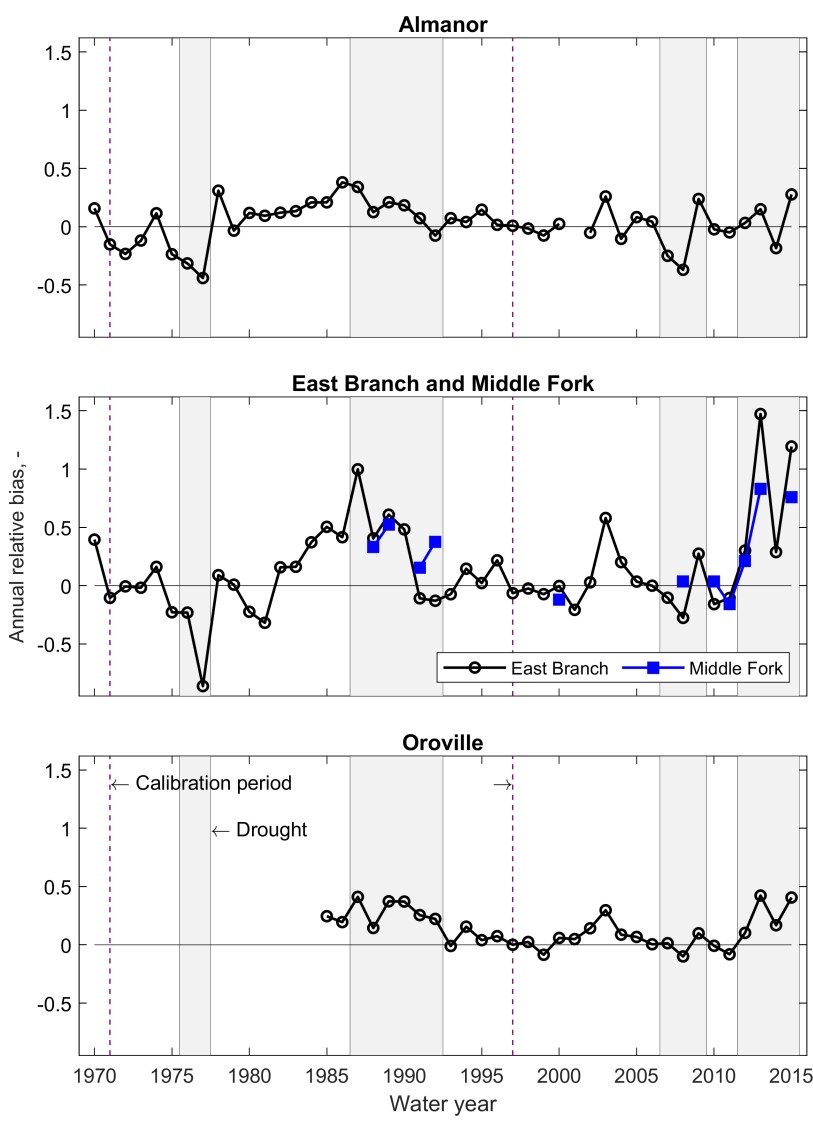

**Figure 6.** Annual relative bias for full-natural flow at the outlet of all (sub-)basins under study.

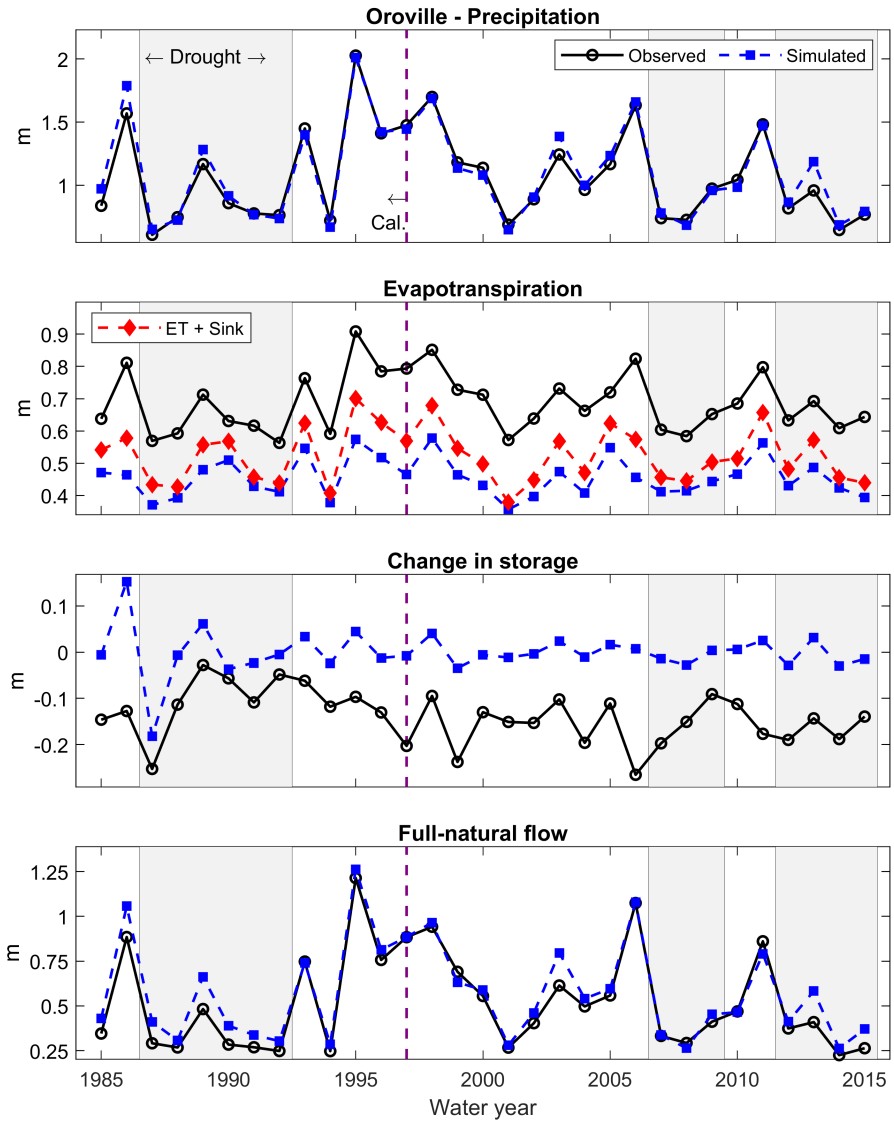

**Figure 7.** Simulated vs. observed annual basin-wide water-balance components ($P$, $ET$, $\Delta S$, and $Q$) for the Feather River at Oroville.



**Figure 8.** Scatter plot of simulated vs. observed annual basin-wide water-balance components ($P$, $ET$, $\Delta S$, and $Q$) separated between drought (red) and non-drought (black) years. Simulated annual $ET$ includes the groundwater-sink mass-flux component (see Section 2.3.3). The red and grey bands represent 95% confidence intervals for the regressions.



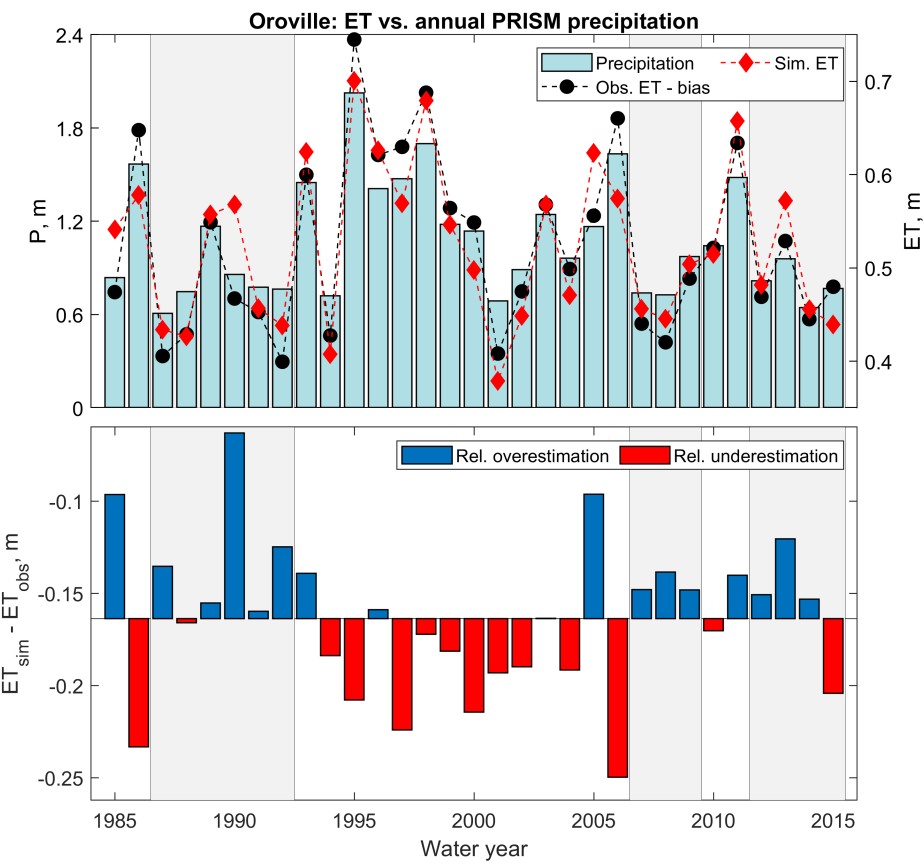

**Figure 9.** Top panel: simulated and observed annual basin-wide evapotranspiration for the Feather River at Oroville (lines) and observed annual precipitation according to PRISM (bar chart). The absolute value of the systematic bias between simulated and observed basin-wide evapotranspiration (∼ 160 mm, see Figure 7) was subtracted from observed values for readability. Bottom panel: annual differences between simulated and observed basin-wide evapotranspiration. Simulated annual evapotranspiration includes the groundwater-sink mass-flux component (see Section 2.3.3).



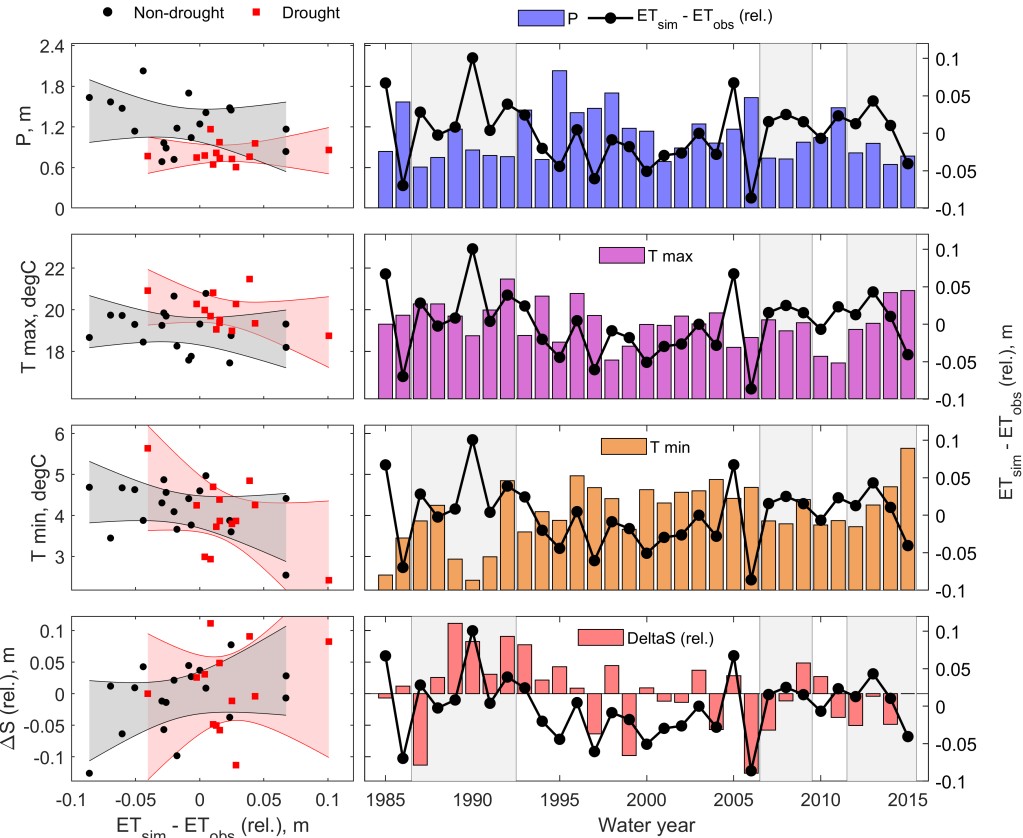

**Figure 10.** Correlation across differences between simulated and observed annual evapotranspiration (relative to the systematic bias, see Figure 7) and four potential predictors (from the top to the bottom): annual PRISM precipitation, annual maximum temperature, annual minimum temperature, and observed annual relative change in storage (also relative to the corresponding systematic bias – Figure S6). Regressions were calculated by separating drought and non-drought years (red and black in left column, respectively). The red and grey bands (left column) represent 95% confidence intervals for the regressions. Maximum and minimum temperature were estimated based on data in Figure 2. Simulated annual evapotranspiration includes the groundwater-sink mass-flux component (see Section 2.3.3).

**Figure 11.** Precipitation-runoff relationship for drought (red) vs. non-drought (black) years and the twelve (main) basins draining the western side of the Sierra Nevada to the California Central Valley in addition to the Feather River. The red and grey bands represent 95% confidence intervals for the regressions.