# Peer review of "Climate elasticity of evapotranspiration shifts the water balance of Mediterranean climates during multi-year droughts"

_Hydrology and Earth System Sciences, 2019_

## Referee Comment (RC1) · Anonymous Referee #1 · 25 Sep 2019

The manuscript by Avanzi et al. is generally well-written and well-referenced. They utilize a hydrologic modeling approach to quantify how a mountain watershed responds to short duration (sub-decadal) drought episodes, and find that how simulated evapotranspiration responds to drought conditions is a major issue with regards to runoff estimation. While unsurprising, this finding is important to motivate improvements in ET in hydrologic models in mountain regions. Overall, the modelling approach is acceptable, but will need some additional information and analysis (noted below), especially with regards to the ET estimation approach and some additional longer simulations.

[Figure]

I am also a bit skeptical of the fact that a groundwater model was not used to show whether or not such a model is necessary to accurately capture hydrologic responses in volcanic, subsurface flow-dominated basins. I liked the shift identification approach for changes in precipitation-runoff relationships. I also thought it was a valuable addition to include a range-wide assessment to highlight not only the application of the approach but also to show how basin response to drought varies as a function of elevation. Some general and specific comments follow below.

General comments: 1. The recent 2012-2015 drought period considered by the authors is inconsistent (P2 L33) with the official declaration period of drought on the website provided (P3 L25), which gives 2012-2016. The modelling exercises will need to be repeated to include water year 2016 if the authors would like to stick to the official declaration of drought. As they do note that the results are sensitive to the duration of drought episodes, it would be worth including (and comparing) the 2012-2015 (what could be called 'peak drought' conditions) and 2012-2016 (the 'official drought') periods. I think it would strengthen the paper to include some additional modeling studies of longer droughts, even if these are hypothetical. Many water agencies (and model experiments) use 'design' droughts based on a few known drought episodes lumped together or informed by past extended droughts. While I see equation 3 was applied by lumping results together on Page 6, I would like to see how a few iterations of a design drought of decadal (or longer) scale change (or do not change) the results.

2. PRMS is a fine modeling approach, but in regions with known groundwater/surface water interactions, such as the volcanic Almanor sub-basin, why was a groundwater model not also developed and coupled to PRMS (i.e., GSFLOW?). I understand this may not improve results, but without showing that it does not, it leaves the reader wondering if such an addition would improve results.

3. P8 L5: The authors misconstrue the definition of 'warm snow drought', perhaps because the definition can be misleading if no thresholds in precipitation anomalies are specified. Though the 2012-2015 drought may have had greater precipitation than

the 1977 drought, many of these years in both drought periods satisfied the dry snow drought conditions demonstrated in Hatchett and McEvoy (2018). Warm snow droughts should only be defined by years with near to above-normal precipitation and below average snowpack.

4. P8 L25: Snow/rain ratios should not be based upon a single point in time value of accumulated precipitation and snowpack at that time (April 1), as this neglects numerous factors that may be controlling the state of the snowpack on this date. For example, if a third of an above-average snowpack was lost during a warm, humid period in late March, the value on April 1 would not correctly represent the fact that otherwise a year had experienced above normal snow and perhaps above normal rain/snow ratios. Just a hypothetical example. I suggest calculating snow fraction over the course of the year and using the total precipitation estimated as snow divided by total precipitation (ending on Apr 1) as a more robust metric.

5. The procedure used to estimate ET is very interesting, but as this method has not yet been accepted by the scientific community (as noted by a submitted paper on P5 L4 and L6), I need to see comparisons of these results with some standard, easily implementable methods to calculate ET.

6. The concept of climate-driven ET elasticity is very cool. I would recommend a schematic figure be added to highlight this concept. The additional model runs using longer drought episodes (warm and dry versus cool and dry at decadal time scales) could play into making this figure more robust by helping constrain the temporal and climate condition sensitivity of the elasticity.

7. The map figures (Figure 1) are not up to publication standards. They need to be projected with latitude and longitude coordinates. The bins of precipitation are far too large and substantial precipitation variability is lost. The inset map needs to be projected and should show the entire west coast of North America (or at least the western United States). I suggest simply binning precipitation by 50 mm increments to better

highlight topographic gradients. The geologic mapping appears to be hand drawn (and is of very poor quality) and needs to be markedly improved. The map should also show the locations of the stations used in the study since they are referenced in the main text.

Specific Comments: P3 L11: Please change all instances of Oroville Lake to the correct title, "Lake Oroville"

P3 L26: When referring to climate, one must not neglect temperature if discussing precipitation. Please change to "dry, hot summers and wet, mild winters".

P3 L28: It would help to add a figure demonstrating the basin hypsometry to quantify 'most'. Does "most" mean 59% or 92%?

P4 L4: Add 'anomalous' before 'low'

P4 L13: Should be "Cascade Range".

P4 L14: Suggest to add more geologic context, specifically on soils.

P4 L22: Please provide the temporal resolutions used. Perhaps a table could be used?

P4 L22: What are the resolutions of the spatially distributed indices?

P4 L29: What types of precipitation gauges were used? Were gauges heated? Are there concerns for undercatch?

P5 L1: Which PRISM products were used, 4 km or 800 m? Monthly or daily?

P5 L25: What alpha level was significance assessed at?

P8 L1: I'm a bit confused here, is the paper referring to flow below Oroville, or total inflow to Oroville? If the former, these numbers need to be prefaced by discussion of water deliveries that may have been subject to changed allocations in response to the drought. Similarly, the storage value in the reservoir doesn't really add much, given that a reservoir's operational goals might be to completely drain the reservoir by the end of

the water year. If additional context is provided, then this number becomes meaningful. Last, please correct 'norm' to the proper spelling 'normal'.

P8 L3: For consistency with the discussion on the 1987-1992 drought on P8 L8, please include the temperature anomaly for the 2012-2016 (note I used the official period there, not 2012-2015).

P8 L9: Section should be singular.

P8 L9: The start of this paragraph is a bit strange, i.e., is there any reason to suspect that runoff seasonality should not be preserved? I think this sentence could be removed or replaced with some more insightful findings. Perhaps discuss any temporal shifts?

P10 L11: Can you add some additional clarity about these winter precipitation events? I would expect these to be extreme precipitation rain-on-snow events to produce peak flows, but as it is written, any precip event could generate a peak flow event.

P10 L30: I am balking at the use of 'observed' water balance, as that implies that the water balance has been completely observed, when in reality it is merely an estimate based upon models for precip (PRISM) and ET (NDVI-GAM approach). "Estimated" might be a better word but could confuse readers against "modeled". I leave this to the authors to ponder if a better descriptor could be used.

P11 L8: What is "soft data"?

P12 L25: Suggest to add a citation for the second and third sources of uncertainty.

P13 L6: I like the concept of tree mortality (despite the myriad complexities controlling tree mortality in addition to ET like pests, disease, etc), but I feel like this sentence detracts from the previous, powerful statement defining climate elasticity of ET. Ending the paragraph with this definition and instead another elucidating sentence about the value of this metric would be a strong way to close out this subsection.

Table 1: Please add the period of record means (or medians) for each variable. These

statistics are not provided in Figure 2 as the caption implies.

---

## Referee Comment (RC2) · Anonymous Referee #2 · 18 Oct 2019

Overall, this is a very interesting and relevant piece of work. However, authors tend to generalize and leave the reader hanging in some cases which raises some questions. Please see below specific comments for your attention.

Title needs modification, it does not communicate the focus of the study, especially after reading the first sentence of the abstract.

**Abstract**
Did authors focus on the effect of drought on shifts in precipitation-runoff relationships and the performance of the model?
Authors need to clarify the catchment/s studies. Or they use one catchment with sub-basins, and if so, the results reported should be for the basins?? The abstract has to be summarized and comprehensive.

**Introduction**
The motivation and novelty of the study is very week. The introduction lacks coherence. For example, the reader has to be able to identify:
1. overall effect of droughts on water balance in Med climate, with specific examples.
2. The approaches of examining precipitation-runoff relationships and how successful they have been in the same climate.
3. What has been done so far in relation to precipitation-runoff relationships or water balance studies within the same climate. This indicate the contribution of the work and its novelty.

**Research questions:** Authors need to improve and rewrite all their research questions to be clear. It's very difficult for the reader to understand them.  They raise a lot of questions: what is the water-balance predictive skill? Are the authors referring to the potential of the model to predict shifts during drought and non-drought periods???
What is the suitability of the generalized additive model in predicting ET for the catchment? Was it used before?

**Method**
What were the characteristics of the Landsat-based annual-averaged NDVI? How was it derived? Is it a product or authors derived their own product through estimation? Any preprocessing of the image?

**Discussion,**
I suggest authors use their main objectives as subheadings for the reader to understand the findings and their implications.

---

## Author Comment (AC1) · 30 Dec 2019

Dear Colleague,

Thank you very much for the thoughtful review of our paper. Please find below our point-by-point reply to comments, including our intended changes to the manuscript. Your comments are in italics, with our replies are in plain text.

*The manuscript by Avanzi et al. is generally well-written and well-referenced. They utilize a hydrologic modeling approach to quantify how a mountain watershed responds*

*to short duration (sub-decadal) drought episodes, and find that how simulated evapotranspiration responds to drought conditions is a major issue with regards to runoff estimation. While unsurprising, this finding is important to motivate improvements in ET in hydrologic models in mountain regions. Overall, the modelling approach is acceptable, but will need some additional information and analysis (noted below), especially with regards to the ET estimation approach and some additional longer simulations.*

*I am also a bit skeptical of the fact that a groundwater model was not used to show whether or not such a model is necessary to accurately capture hydrologic responses in volcanic, subsurface flow-dominated basins. I liked the shift identification approach for changes in precipitation-runoff relationships. I also thought it was a valuable addition to include a range-wide assessment to highlight not only the application of the approach but also to show how basin response to drought varies as a function of elevation.*

We agree that our results motivate improvements in ET parametrizations, especially since the observed shifts extend to the majority of water basins in the Sierra Nevada and occur regardless of predominant geology. We also agree that coupling a groundwater model to PRMS could shed further light on shifts in the basin water balance, and potentially precipitation-runoff relationships. While a coupled version of PRMS is available (GSFLOW), setting it up with the same level of spatial data and parameterization as it was done for PRMS would be a major new study, requiring an effort that goes well beyond the scope of this manuscript. Lacking the data needed to describe groundwater flow in the basin with a high level of rigor, it is our assessment that the simplified simulation of groundwater processes in PRMS is appropriate to meet the aims of this study. In this simplified setting, PRMS and many other hydrologic models with similar process representations are currently being used by water and energy forecasters in California and elsewhere to predict water supply at various time scales. Highlighting that such hydrologic models are prone to performance drops during shifts in precipitation-runoff relationships thus addresses an important aspect of operational hydrology. We will certainly add more discussion on this point in our paper, including

suggestions to perform coupled experiments in the future.

*Some general and specific comments follow below.*

*General comments: 1. The recent 2012-2015 drought period considered by the authors is inconsistent (P2 L33) with the official declaration period of drought on the website provided (P3 L25), which gives 2012-2016. The modelling exercises will need to be repeated to include water year 2016 if the authors would like to stick to the official declaration of drought. As they do note that the results are sensitive to the duration of drought episodes, it would be worth including (and comparing) the 2012-2015 (what could be called 'peak drought' conditions) and 2012-2016 (the 'official drought') periods.*

We agree to extend the analysis to the 2012-2016 period. While WY 2016 had average precipitation, including it may give further insight into the drought response. Note, however, that there are multiple definitions of drought, and the one on the DWR web page refers to impacts on water users.

*I think it would strengthen the paper to include some additional modeling studies of longer droughts, even if these are hypothetical. Many water agencies (and model experiments) use 'design' droughts based on a few known drought episodes lumped together or informed by past extended droughts. While I see equation 3 was applied by lumping results together on Page 6, I would like to see how a few iterations of a design drought of decadal (or longer) scale change (or do not change) the results.*

We agree that scenarios involving longer droughts can inform water management, and hope that our results can improve those studies. As to using scenarios in the current study, that would involve developing different data sets that would have less certainty in assessing model response. All results in this paper rely on measurements, including detecting shifts in precipitation vs. runoff, assessing the performance of the PRMS model during droughts and wet periods, and comparing estimated and modeled water-balance components. While paleoclimatic datasets suggest prolonged, multidecadal droughts in California, it would thus be challenging to generate the observational dataset we need to fully apply our methods. In addition, drought vs. non-drought conditions in California have a strong interannual character because of the quasiperiodicity of El Nino–Southern Oscillation (https://doi.org/10.1002/2014GL062433), meaning that investigating these shorter time scales is functional to answering our research questions. We will add more discussion on this point.

*2. PRMS is a fine modeling approach, but in regions with known groundwater/surface water interactions, such as the volcanic Almanor sub-basin, why was a groundwater model not also developed and coupled to PRMS (i.e., GSFLOW?). I understand this may not improve results, but without showing that it does not, it leaves the reader wondering if such an addition would improve results.*

Please see response to first general comment, above.

*3. P8 L5: The authors misconstrue the definition of 'warm snow drought', perhaps because the definition can be misleading if no thresholds in precipitation anomalies are specified. Though the 2012-2015 drought may have had greater precipitation than the 1977 drought, many of these years in both drought periods satisfied the dry snow drought conditions demonstrated in Hatchett and McEvoy (2018). Warm snow droughts should only be defined by years with near to above-normal precipitation and below average snowpack.*

We agree and will clarify this point in the manuscript.

*4. P8 L25: Snow/rain ratios should not be based upon a single point in time value of accumulated precipitation and snowpack at that time (April 1), as this neglects numerous factors that may be controlling the state of the snowpack on this date. For example, if a third of an above-average snowpack was lost during a warm, humid period in late March, the value on April 1 would not correctly represent the fact that otherwise a year had experienced above normal snow and perhaps above normal rain/snow ratios. Just a hypothetical example. I suggest calculating snow fraction over the course of the year*

*and using the total precipitation estimated as snow divided by total precipitation (ending on Apr 1) as a more robust metric.*

We agree with this suggestion and will try to recalculate results in Section 3.1 and Figure 2 using the suggested metric. A caveat here is that data reported in Figure 2 are monthly, and daily data may be not publicly available at some of the precipitation stations considered. Also, some of these precipitation data may lack co-located air-temperature (and optionally relative-humidity) data to estimate phase partitioning between rain and snow, and vice versa. During our revision, we will assess data availability from this standpoint and will propose our best estimate.

*5. The procedure used to estimate ET is very interesting, but as this method has not yet been accepted by the scientific community (as noted by a submitted paper on P5 L4 and L6), I need to see comparisons of these results with some standard, easily implementable methods to calculate ET.*

The procedure is a modification of the well-cited approach by Goulden et al. (2012), which has been used in multiple papers since then (see references to our work, below). The Rungee manuscript mentioned on P5 L4 and L6 will be submitted to HESS by R. Bales (Rungee has moved on to a new position), and if accepted as a discussion paper all methods related to the updated ET product we used here will be freely accessible online by the time the revised version of this manuscript will be submitted. The authors welcome public comments on the method in the Rungee manuscript, which moves from one independent variable (NDVI) in published papers by Goulden/Bales to 2 independent variables (NDVI and precipitation). Adding the additional variable both recognizes the responsiveness of ET to precipitation, independent of NDVI, and provides a lower error. The accuracy of this ET product will be also discussed in the revised version of this manuscript, and we will include an analysis of the 2-variable vs 1-variable approach in the supplement as needed. There is no independent, accurate spatial product for ET for the Feather River basin; however, we present a leave-one-out cross validation in the Rungee paper. Finally, we are also creating a DOI for the

flux-tower datasets and ET products in that paper, which are published in the Rungee (2018) paper cited in our manuscript.

References

1. Goulden, M.L., R.G. Anderson, R.C. Bales, A.E. Kelly, M. Meadows and G.C. Winston, "Evapotranspiration along an Elevation Gradient in California's Sierra Nevada," Journal of Geophysical Research, September 2012, Vol. 117, G03028.

2. Goulden, M.L. and R.C. Bales, "Mountain Runoff Vulnerability to Increased Evapotranspiration with Vegetation Expansion," Proceedings of the National Academy of Sciences of the United States of America, September 2014, Vol. 111, No. 39, pp. 14071-14075.

3. Goulden, M.L. Bales, R.C. California forest die-off linked to multi-year deep soil drying in 2012–2015 drought, Nature Geoscience 12,632–637 (2019)

4. J.W. Roche, M.L. Goulden, R.C. Bales. Estimating evapotranspiration change due to forest treatment and fire at the basin scale in the Sierra Nevada, California. Ecohydrol., 2018

5. R.C. Bales, M.L. Goulden. C.T. Hunsaker, M.H. Conklin, P.C. Hartsoug, A.T. O'Geen, J.W. Hopmans, M. Safeeq. Mechanisms controlling the impact of multi-year drought on mountain hydrology, Scientific Reports, 2018.

6. A.W. Fellows | M. L. Goulden. Mapping and understanding dry season soil water drawdown by California montane vegetation, Ecohydrol.2017

*6. The concept of climate-driven ET elasticity is very cool. I would recommend a schematic figure be added to highlight this concept. The additional model runs using longer drought episodes (warm and dry versus cool and dry at decadal time scales) could play into making this figure more robust by helping constrain the temporal and climate condition sensitivity of the elasticity.*

[Figure]

We will work on a schematic as suggested.

*7. The map figures (Figure 1) are not up to publication standards. They need to be projected with latitude and longitude coordinates. The bins of precipitation are far too large and substantial precipitation variability is lost. The inset map needs to be projected and should show the entire west coast of North America (or at least the western United States). I suggest simply binning precipitation by 50 mm increments to better highlight topographic gradients. The geologic mapping appears to be hand drawn (and is of very poor quality) and needs to be markedly improved. The map should also show the locations of the stations used in the study since they are referenced in the main text.*

We will revise Figure 1 as suggested. The geological map was not hand drawn and is an adaption of the USGS National Atlas in which groups were aggregated based on the main geology (volcanic, sedimentary, granitic).

*Specific Comments: P3 L11: Please change all instances of Oroville Lake to the correct title, "Lake Oroville"*

*P3 L26: When referring to climate, one must not neglect temperature if discussing precipitation. Please change to "dry, hot summers and wet, mild winters".*

*P3 L28: It would help to add a figure demonstrating the basin hypsometry to quantify 'most'. Does "most" mean 59% or 92%?*

*P4 L4: Add 'anomalous' before 'low'*

*P4 L13: Should be "Cascade Range".*

*P4 L14: Suggest to add more geologic context, specifically on soils.*

We will incorporate all comments above in the revised manuscript.

*P4 L22: Please provide the temporal resolutions used. Perhaps a table could be used?*
We agree and will add a Table to the Supporting Information summarizing details about the temporal-spatial resolutions of all data used.

*P4 L22: What are the resolutions of the spatially distributed indices?*

The temporal resolution is annual (see P4 L22). The spatial resolution is discussed on P5 L9-10. We will include this resolution on P4 L22 too.

*P4 L29: What types of precipitation gauges were used? Were gauges heated? Are there concerns for undercatch?*

Precipitation gauges used in the river basin are managed by various agencies, with the California Department of Water Resources collecting and archiving the data. We will add the responsible agency to Table S1. The design of these sensors resembles the one in use by the SNOTEL network throughout the western US (https://www.wcc.nrcs.usda.gov/about/mon_automate.html). Most gauges are unheated and some are manually measured by on-site agency personnel. Most are also located in small clearings where wind speed is low, which suggests that undercatch is locally low, especially below the seasonal rain-snow line. Nevertheless, we agree that undercatch increases in snow-dominated regions and will add all these details, plus a discussion of the effect of undercatch on our results, to the manuscript.

*P5 L1: Which PRISM products were used, 4 km or 800 m? Monthly or daily?*

The spatial resolution is reported on P5 L9 (800 m), while the temporal resolution was annual (see on P4 L22). Note that PRISM data in this paper were also used through the DRAPER approach to estimate precipitation input data for PRMS (see on P5 L29ff). For this second scope, we used monthly polygon surfaces of precipitation contours (P5 L29). All this information will be summarized in the Supporting Information.

*P5 L25: What alpha level was significance assessed at?*

This information is reported on P9 L19 but we will include it at P5 L25 as well.
*P8 L1: I'm a bit confused here, is the paper referring to flow below Oroville, or total inflow to Oroville? If the former, these numbers need to be prefaced by discussion of water deliveries that may have been subject to changed allocations in response to the drought. Similarly, the storage value in the reservoir doesn't really add much, given that a reservoir's operational goals might be to completely drain the reservoir by the end of the water year. If additional context is provided, then this number becomes meaningful. Last, please correct 'norm' to the proper spelling 'normal'.*

Based on our understanding of the technical report cited at line 2 (DWR, 1978), this is full-natural flow at Lake Oroville, which is comparable to inflow to the reservoir and is independent from water-allocation decisions downstream of the dam. It is true that reservoir storage is highly seasonal, but it also responds to multiple objectives that may require reservoir level to be maintained to a certain high level (e.g., recreational reasons, hydropower production, multi-year carryover for water supply, etc). We will add this context as requested and also correct the word 'normal'.

*P8 L3: For consistency with the discussion on the 1987-1992 drought on P8 L8, please include the temperature anomaly for the 2012-2016 (note I used the official period there, not 2012-2015)*

*P8 L9: Section should be singular.*

We will include all these corrections in the revised manuscript.

*P8 L9: The start of this paragraph is a bit strange, i.e., is there any reason to suspect that runoff seasonality should not be preserved? I think this sentence could be removed or replaced with some more insightful findings. Perhaps discuss any temporal shifts?*

Runoff seasonality depends on snow-rain proportion and ultimately on the timing of precipitation input to the system, both of which may be affected by droughts. Therefore, runoff seasonality may in principle not be preserved. We will revise this paragraph to clarify this point; we will also discuss temporal shifts as kindly suggested.

[Figure]

*P10 L11: Can you add some additional clarity about these winter precipitation events? I would expect these to be extreme precipitation rain-on-snow events to produce peak flows, but as it is written, any precip event could generate a peak flow event.*

The Feather river lies across the seasonal rain-snow transition zone. As such, rain-on-snow as well as mixed rain-snow events are frequent. These events significantly increase streamflow compared to periods with no precipitation, as typical hydrographs on the river show (see https://pubs.usgs.gov/sir/2004/5202/sir2004-5202.pdf, page 27). In the revised manuscript, we will include some of these hydrographs in the Supporting Information to clarify this point.

*P10 L30: I am balking at the use of 'observed' water balance, as that implies that the water balance has been completely observed, when in reality it is merely an estimate based upon models for precip (PRISM) and ET (NDVI-GAM approach). "Estimated" might be a better word but could confuse readers against "modeled". I leave this to the authors to ponder if a better descriptor could be used*

We agree and will use 'estimated' in the revised manuscript. In so doing, we will add one sentence in Section 2.3.3 to clearly define what we mean with 'estimated' and 'modeled'.

*P11 L8: What is "soft data"?*

Measuring sub-surface-storage decline is challenging, meaning one has to rely on indirect observations (e.g., magnitude of low flows or rate of seasonal flow from springs). Some of these indirect observations are discussed in Freeman (2011). This was our intended meaning of 'soft data', but we will revise the manuscript using some of the wording we used here to clarify this point.

*P12 L25: Suggest to add a citation for the second and third sources of uncertainty.*

Bales et al. (2018), which is cited at the very beginning of this paragraph, is the main citation for all the sources of uncertainty in this paragraph. We will clarify this point.
*P13 L6: I like the concept of tree mortality (despite the myriad complexities controlling tree mortality in addition to ET like pests, disease, etc), but I feel like this sentence detracts from the previous, powerful statement defining climate elasticity of ET. Ending the paragraph with this definition and instead another elucidating sentence about the value of this metric would be a strong way to close out this subsection.*

We will edit the section as suggested.

*Table 1: Please add the period of record means (or medians) for each variable. These statistics are not provided in Figure 2 as the caption implies.*

We will add the period of record as suggested. Figure 2 reports annual average maximum and minimum temperature, annual quartiles of cumulative precipitation and April1 SWE, and annual April1 SWE / cumulative precipitation. We aggregated these values to obtain statistics in Table 1. This was our intended meaning of caption in Table 1. This will be revised.
* * *

---

## Author Comment (AC2) · 30 Dec 2019

Dear Colleague,

Thank you very much for the thoughtful review of our paper. Please find below our point-by-point reply to comments, including our intended changes to the manuscript. Your comments are in italics, with our replies are in plain text.

*Overall, this is a very interesting and relevant piece of work. However, authors tend to generalize and leave the reader hanging in some cases which raises some questions.*

[Figure]

*Please see below specific comments for your attention.*

Thank you for your comments, which will be considered in our revision.

*Title needs modification, it does not communicate the focus of the study, especially after reading the first sentence of the abstract.*

Our choice was to summarize the main finding of this paper as title. Albeit infrequent, this choice is increasingly popular in hydrologic literature (https://www.the-cryosphere.net/8/257/2014/, https://www.pnas.org/content/110/38/15216, https://agupubs.onlinelibrary.wiley.com/doi/10.1002/2016GL071999, just to mention a few) and it does, in our opinion, communicate what is the main focus of the study, that is, the impact of droughts on the water balance of Mediterranean mixed rain-snow catchments.

At the same time, we acknowledge that some word choices in the current title may sound unfamiliar for the broad audience (e.g., precipitation-runoff relationships). We will therefore propose a revised title as follows: 'Evapotranspiration feedbacks during multi-year droughts shift the water balance of mixed rain-snow catchments'

*Abstract*

*Did authors focus on the effect of drought on shifts in precipitation-runoff relationships and the performance of the model? Authors need to clarify the catchment/s studies. Or they use one catchment with sub-basins, and if so, the results reported should be for the basins?? The abstract has to be summarized and comprehensive.*

We focused on three points: (1) we quantified shifts in precipitation-runoff relationships in four nested catchments of the Feather River; (2) we assessed performances of the PRMS model in these nested catchments during droughts and in particular during periods corresponding to shifts in the water balance; (3) by leveraging the fact that the performance of the model was sensitive to these shifts, we identified the water-balance term for which accuracy during droughts was statistically different from accuracy dur-

ing wet periods (ET). We concluded that a different functioning of ET between droughts and non-drought periods is the likely cause of these shifts.

This point is general and goes beyond the specific catchments we considered. In the current version of the abstract we thus focused on these general points. In the revised manuscript, we will add more details on results of the nested-catchment studies (e.g., differences between sub-surface-dominated and surface-runoff dominated catchments).

*Introduction The motivation and novelty of the study is very week. The introduction lacks coherence. For example, the reader has to be able to identify:*

*1. overall effect of droughts on water balance in Med climate, with specific examples.*

*2. The approaches of examining precipitation-runoff relationships and how successful they have been in the same climate.*

*3. What has been done so far in relation to precipitation-runoff relationships or water balance studies within the same climate. This indicate the contribution of the work and its novelty.*

We will fully revise the Introduction following reviewers' suggestions. We agree that the Introduction must focus more on droughts in regions with a Mediterranean climate.

*Research questions: Authors need to improve and rewrite all their research questions to be clear. It's very difficult for the reader to understand them. They raise a lot of questions: what is the water-balance predictive skill? Are the authors referring to the potential of the model to predict shifts during drought and non-drought periods???*

We will revise all questions for clarity. In so doing, we will avoid any wording that could sound too specific and unsuitable for a broader audience.

With 'water-balance predictive skill', we meant the performance of PRMS in predicting precipitation, ET, changes in sub-surface storage, and runoff. This will be also clarified.
*What is the suitability of the generalized additive model in predicting ET for the catchment? Was it used before?*

The procedure is a modification of the well-cited approach by Goulden et al. (2012), which has been used in multiple papers since then (see references to our work, below). The Rungee manuscript mentioned on P5 L4 and L6 will be submitted to HESS by R. Bales (Rungee has moved on to a new position), and if accepted as a discussion paper all methods related to the updated ET product we used here will be freely accessible online by the time the revised version of this manuscript will be submitted. The authors welcome public comments on the method in the Rungee manuscript, which moves from one independent variable (NDVI) in published papers by Goulden/Bales to 2 independent variables (NDVI and precipitation). Adding the additional variable both recognizes the responsiveness of ET to precipitation, independent of NDVI, and provides a lower error. The accuracy of this ET product will be also discussed in the revised version of this manuscript, and we will include an analysis of the 2-variable vs 1-variable approach in the supplement as needed. There is no independent, accurate spatial product for ET for the Feather River basin; however, we present a leave-one-out cross validation in the Rungee paper. Finally, we are also creating a DOI for the flux-tower datasets and ET products in that paper, which are published in the Rungee (2018) paper cited in our manuscript.

References

1. Goulden, M.L., R.G. Anderson, R.C. Bales, A.E. Kelly, M. Meadows and G.C. Winston, "Evapotranspiration along an Elevation Gradient in California's Sierra Nevada," Journal of Geophysical Research, September 2012, Vol. 117, G03028.

2. Goulden, M.L. and R.C. Bales, "Mountain Runoff Vulnerability to Increased Evapotranspiration with Vegetation Expansion," Proceedings of the National Academy of Sciences of the United States of America, September 2014, Vol. 111, No. 39, pp. 14071-14075.

3. Goulden, M.L. Bales, R.C. California forest die-off linked to multi-year deep soil drying in 2012–2015 drought, Nature Geoscience 12,632–637 (2019)

4. J.W. Roche, M.L. Goulden, R.C. Bales. Estimating evapotranspiration change due to forest treatment and fire at the basin scale in the Sierra Nevada, California. Ecohydrol., 2018

5. R.C. Bales, M.L. Goulden. C.T. Hunsaker, M.H. Conklin, P.C. Hartsoug, A.T. O'Geen, J.W. Hopmans, M. Safeeq. Mechanisms controlling the impact of multi-year drought on mountain hydrology, Scientific Reports, 2018.

6. A.W. Fellows | M. L. Goulden. Mapping and understanding dry season soil water drawdown by California montane vegetation, Ecohydrol.2017

*Method*

*What were the characteristics of the Landsat-based annual-averaged NDVI? How was it derived? Is it a product or authors derived their own product through estimation? Any preprocessing of the image?*

Landsat 5, 7 and 8 were used to map NDVI at 30-m resolution. Values were calculated from the Tier 1 surface-reflectance product downloaded from Google Earth Engine. NDVI values among different Landsat sensors were homogenized by cross-calibrating Landsat 7 (NDVI in 2012) and Landsat 8 (NDVI in 2013-2016) into Landsat 5. Annual Landsat NDVI maps were generated by averaging all pixels in a water year. Pixels with shadow, snow, or cloud were excluded from the calculation. We thought this too much detail for the manuscript, but can add it as needed.

*Discussion,*

*I suggest authors use their main objectives as subheadings for the reader to understand the findings and their implications.*

The three subsections of the Discussion do not directly relate to the main objectives

stated in the Introduction, but look at results from other perspectives to further clarify their implications. The main results of the paper and their link to our main objectives will be clarified in the first paragraph of the discussion.

---

## Author Response (AR1)

Savona (Italy)

May 16, 2020

Dear Editor,

We would like to submit the manuscript *Climate elasticity of evapotranspiration during multi-year droughts shifts the water balance of Mediterranean rain-snow climates* for publication in Hydrology and Earth System Sciences. This is a resubmission of manuscript **hess-2019-377**, which was published in open discussion on August 26, 2019 and was reviewed by two referees. In doing so, we would like to thank the Editor and the Editorial Office for their patience and flexibility with our deadlines.

We have revised and improved the manuscript based on comments from all reviewers and would like to thank all of you for taking the time to review our manuscript. Our revisions followed the expected changes we outlined in our public response to referees. We prioritized (1) providing more details about the spatially distributed evapotranspiration product we used in this paper, (2) improving our Introduction to better frame our research questions, and (3) expanding our discussion to clarify climate elasticity of evapotranspiration, including a schematic as suggested by Referee 1. We also included in the manuscript some discussion on why we did not consider a groundwater model or a hypothetical design drought, again following the arguments we provided in our public response. All minor points were fixed following comments of both referees.

Please find attached our point-by-point replies and the new version of our manuscript for details. We also attached a version of the manuscript with tracked changes.

*Francesco Avanzi and coauthors*

**Reply to Reviewer #1**

The manuscript by Avanzi et al. is generally well-written and well-referenced. They utilize a hydrologic modeling approach to quantify how a mountain watershed responds to short duration (sub-decadal) drought episodes, and find that how simulated evapotranspiration responds to drought conditions is a major issue with regards to runoff estimation. While unsurprising, this finding is important to motivate improvements in ET in hydrologic models in mountain regions. Overall, the modelling approach is acceptable, but will need some additional information and analysis (noted below), especially with regards to the ET estimation approach and some additional longer simulations.

I am also a bit skeptical of the fact that a groundwater model was not used to show whether or not such a model is necessary to accurately capture hydrologic responses in volcanic, subsurface flow-dominated basins. I liked the shift identification approach for changes in precipitation-runoff relationships. I also thought it was a valuable addition to include a range-wide assessment to highlight not only the application of the approach but also to show how basin response to drought varies as a function of elevation.

> We agree that our results motivate improvements in $ET$ parametrizations, especially since the observed shifts extend to the majority of water basins in the Sierra Nevada and occur regardless of predominant geology.

General comments: 1. The recent 2012-2015 drought period considered by the authors is inconsistent (P2 L33) with the official declaration period of drought on the website provided (P3 L25), which gives 2012-2016. The modelling exercises will need to be repeated to include water year 2016 if the authors would like to stick to the official declaration of drought. As they do note that the results are sensitive to the duration of drought episodes, it would be worth including (and comparing) the 2012-2015 (what could be called 'peak drought' conditions) and 2012-2016 (the 'official drought') periods.

> Changes: We included water year 2016 to all results in this manuscript. We did not add a specific discussion on this addition as this was considered a little out of scope for this study.

I think it would strengthen the paper to include some additional modeling studies of longer droughts, even if these are hypothetical. Many water agencies (and model experiments) use 'design' droughts based on a few known drought episodes lumped together or informed by past extended droughts. While I see equation 3 was applied by lumping results together on Page 6, I would like to see how a few iterations of a design drought of decadal (or longer) scale change (or do not change) the results.

> We agree that scenarios involving longer droughts can inform water management, and hope that our results can improve those studies. As to using scenarios in the current study, that would involve developing different data sets that would have less certainty in assessing model response. All results in this paper rely on measurements, including detecting shifts in precipitation vs. runoff, assessing the performance of the PRMS model during droughts and wet periods, and comparing estimated and modeled water-balance components. While paleoclimatic datasets suggest prolonged, multi-decadal droughts in California, it would thus

be challenging to generate the observational dataset we need to fully apply our methods. In addition, drought vs. non-drought conditions in California have a strong interannual character because of the quasiperiodicity of El Nino–Southern Oscillation [8], meaning that investigating these shorter time scales is functional to answering our research questions.

Changes: we added all points above to the manuscript (see lines 24ff page 7).

**2. PRMS is a fine modeling approach, but in regions with known groundwater/surface water interactions, such as the volcanic Almanor sub-basin, why was a groundwater model not also developed and coupled to PRMS (i.e., GSFLOW?). I understand this may not improve results, but without showing that it does not, it leaves the reader wondering if such an addition would improve results.**

We agree that coupling a groundwater model to PRMS could shed further light on shifts in the basin water balance, and potentially precipitation-runoff relationships. While a coupled version of PRMS is available (GSFLOW), setting it up with the same level of spatial data and parameterization as it was done for PRMS would be a major new study, requiring an effort that goes well beyond the scope of the research presented in this manuscript. Lacking the data needed to describe groundwater flow in the basin with a high level of rigor, it is our assessment that the simplified simulation of groundwater processes in PRMS is appropriate to meet the aims of this study. In this simplified setting, PRMS and many other hydrologic models with similar process representations are currently being used by water and energy forecasters in California and elsewhere to predict water supply at various time scales. Highlighting that such hydrologic models are prone to performance drops during shifts in precipitation-runoff relationships thus addresses an important aspect of operational hydrology.

Changes: we added a brief discussion on this point in our paper (see lines 13ff page 8).

**3. P8 L5: The authors misconstrue the definition of 'warm snow drought', perhaps because the definition can be misleading if no thresholds in precipitation anomalies are specified. Though the 2012-2015 drought may have had greater precipitation than the 1977 drought, many of these years in both drought periods satisfied the dry snow drought conditions demonstrated in Hatchett and McEvoy (2018). Warm snow droughts should only be defined by years with near to above-normal precipitation and below average snowpack.**

Changes: we removed any reference to snow droughts in the manuscript, since these processes were not the focus of the present study.

**4. P8 L25: Snow/rain ratios should not be based upon a single point in time value of accumulated precipitation and snowpack at that time (April 1), as this neglects numerous factors that may be controlling the state of the snowpack on this date. For example, if a third of an above-average snowpack was lost during a warm, humid period in late March, the value on April 1 would not correctly represent the fact that otherwise a year had experienced above normal snow and perhaps above normal rain/snow ratios. Just a hypothetical example. I suggest calculating snow fraction over the course of the year and using the total precipitation estimated as snow divided by total precipitation (ending on Apr 1) as a more robust metric.**

We considered to add the above metric and assessed that its computation would require a fairly large amount of assumptions since data reported in Figure 2 are monthly. Also, some

of these precipitation data lack co-located air-temperature (and optionally relative-humidity) data to estimate phase partitioning between rain and snow.

Changes: we removed snow/rain ratios from Figure 2 in the manuscript. These only had an illustrative purpose and were not necessary to answer our research questions.

**5. The procedure used to estimate ET is very interesting, but as this method has not yet been accepted by the scientific community (as noted by a submitted paper on P5 L4 and L6), I need to see comparisons of these results with some standard, easily implementable methods to calculate ET.**

The procedure is a modification of the well-cited approach by [5], which has been used in multiple papers since then [6, 3, 1, 7]. The main change compared to [5] is the addition of precipitation as predictor, which both recognizes the responsiveness of $ET$ to precipitation, independent of NDVI, and provides a lower error. Further details can be found in [12].

Changes: we added further details about the procedure used to estimate ET in the main text **(see lines 1ff page 6)**, and also point the reader to the recently published paper by [12] – see an author formatted, accepted version at `https://escholarship.org/uc/item/4kb947md`.

**6. The concept of climate-driven ET elasticity is very cool. I would recommend a schematic figure be added to highlight this concept. The additional model runs using longer drought episodes (warm and dry versus cool and dry at decadal time scales) could play into making this figure more robust by helping constrain the temporal and climate condition sensitivity of the elasticity.**

Changes: we added a schematic to the manuscript as suggested **(see lines 3ff page 15)**.

**7. The map figures (Figure 1) are not up to publication standards. They need to be projected with latitude and longitude coordinates. The bins of precipitation are far too large and substantial precipitation variability is lost. The inset map needs to be projected and should show the entire west coast of North America (or at least the western United States). I suggest simply binning precipitation by 50 mm increments to better highlight topographic gradients. The geologic mapping appears to be hand drawn (and is of very poor quality) and needs to be markedly improved. The map should also show the locations of the stations used in the study since they are referenced in the main text.**

Changes: we modified Figure 1 following these suggestions, with only two exceptions: (1) we kept a fairly small number of bins in the precipitation map because this allowed us to highlight the sharp transition between the wet, western side of the basin and the dry, eastern side; this is the main point we would like to convey there; (2) the geological map was removed and we pointed to publicly available reports where such maps are available [9].

**Specific Comments: P3 L11: Please change all instances of Oroville Lake to the correct title, "Lake Oroville"**

Changes: done.

**P3 L26: When referring to climate, one must not neglect temperature if discussing precipitation. Please change to "dry, hot summers and wet, mild winters".**

Changes: done.

**P3 L28: It would help to add a figure demonstrating the basin hypsometry to quantify 'most'. Does "most" mean 59% or 92%?**

Changes: done (**see lines 16ff page 4**).

**P4 L4: Add 'anomalous' before 'low'**

Changes: done.

**P4 L13: Should be "Cascade Range".**

Changes: done.

**P4 L14: Suggest to add more geologic context, specifically on soils.**

Changes: done (**see lines 1ff page 5**).

**P4 L22: Please provide the temporal resolutions used. Perhaps a table could be used?**

Changes: We added a table in the Supporting Information detailing all datasets used, their temporal-spatial resolution, and their role in our study (**see Table S1**).

**P4 L22: What are the resolutions of the spatially distributed indices?**

Changes: We added a table in the Supporting Information detailing all datasets used, their temporal-spatial resolution, and their role in our study (**see Table S1**).

**P4 L29: What types of precipitation gauges were used? Were gauges heated? Are there concerns for undercatch?**

Changes: We added a paragraph in the manuscript to discuss all these points – thank you (**see lines 27ff page 5**).

**P5 L1: Which PRISM products were used, 4 km or 800 m? Monthly or daily?**

Changes: We added a table in the Supporting Information detailing all datasets used, their temporal-spatial resolution, and their role in our study (**see Table S1**). Same information was added in **Section 2.2** for completeness.

**P5 L25: What alpha level was significance assessed at?**

Changes: significance level $\alpha = 5\%$ (**see line 1 page 7**).

**P8 L1: I'm a bit confused here, is the paper referring to flow below Oroville, or total inflow to Oroville? If the former, these numbers need to be prefaced by discussion of water deliveries that may have been subject to changed allocations in response to the drought. Similarly, the storage value in the reservoir doesn't really add much, given that a reservoir's operational goals might be to completely drain the reservoir by the end of the water year. If**

**additional context is provided, then this number becomes meaningful. Last, please correct 'norm' to the proper spelling 'normal'.**

> Based on our understanding of the technical report by DWR [2], this is full-natural flow at Oroville, which is comparable to inflow to the reservoir and is independent from water-allocation decisions downstream of the dam. It is true that reservoir storage is highly seasonal, but it also responds to multiple objectives that may require reservoir level to be maintained to a certain high level (e.g., recreational reasons, hydropower production, multi-year carryover for water supply, etc).
> Changes: no change needed, besides removing the word 'normal'.

**P8 L3: For consistency with the discussion on the 1987-1992 drought on P8 L8, please include the temperature anomaly for the 2012-2016 (note I used the official period there, not 2012-2015). P8 L9: Section should be singular.**

> Changes: done.

**P8 L9: The start of this paragraph is a bit strange, i.e., is there any reason to suspect that runoff seasonality should not be preserved? I think this sentence could be removed or replaced with some more insightful findings. Perhaps discuss any temporal shifts?**

> Changes: we revised the paragraph as suggested (see lines 23ff page 10).

**P10 L11: Can you add some additional clarity about these winter precipitation events? I would expect these to be extreme precipitation rain-on-snow events to produce peak flows, but as it is written, any precip event could generate a peak flow event.**

> Changes: we revised the paragraph as suggested (see lines 24ff page 11).

**P10 L30: I am balking at the use of 'observed' water balance, as that implies that the water balance has been completely observed, when in reality it is merely an estimate based upon models for precip (PRISM) and ET (NDVI-GAM approach). "Estimated" might be a better word but could confuse readers against "modeled". I leave this to the authors to ponder if a better descriptor could be used.**

> Changes: wherever possible, we used the word *estimated* to clarify that none of those water fluxes were directly measured, but are the results of statistical models. We also specified that in Section 2.3.3. and all results thereof, the word *observed* is used as opposed to PRMS-*modeled* fluxes (see lines 24ff page 9).

**P11 L8: What is "soft data"?**

> Measuring sub-surface-storage decline is challenging, meaning one has to rely on indirect observations (e.g., magnitude of low flows or rate of seasonal flow from springs). Some of these indirect observations are discussed in [4]. This was our intended meaning of 'soft data'.
> Changes: we revised the paragraph using some of the wording we used here to clarify this point (see lines 17ff page 12).

**P12 L25: Suggest to add a citation for the second and third sources of uncertainty.**

[1], which is cited at the very beginning of this paragraph, is the main citation for all the sources of uncertainty in this paragraph. No change needed.

**P13 L6: I like the concept of tree mortality (despite the myriad complexities controlling tree mortality in addition to ET like pests, disease, etc), but I feel like this sentence detracts from the previous, powerful statement defining climate elasticity of ET. Ending the paragraph with this definition and instead another elucidating sentence about the value of this metric would be a strong way to close out this subsection.**

Changes: we rephrased the paragraph as suggested.

**Table 1: Please add the period of record means (or medians) for each variable. These statistics are not provided in Figure 2 as the caption implies.**

Figure 2 reports annual average maximum and minimum temperature, and annual quartiles of cumulative precipitation and April-1 SWE. We aggregated these values to obtain statistics in Table 1.

Changes: caption in Table 1 was revised accordingly.

**Reply to Reviewer #2**

**Overall, this is a very interesting and relevant piece of work. However, authors tend to generalize and leave the reader hanging in some cases which raises some questions. Please see below specific comments for your attention.**

**Title needs modification, it does not communicate the focus of the study, especially after reading the first sentence of the abstract.**

> Our choice was to summarize the main take-away of this paper as title. Albeit infrequent, this choice is increasingly popular in hydrologic literature [13, 10, 11] and it does, in our opinion, communicate what is the main focus of the study, that is, how the buffered response of $ET$ to precipitation variability impacts the water balance of Mediterranean mixed rain-snow catchments. At the same time, we acknowledge that some word choices in the current title may sound unfamiliar for the broad audience (e.g., precipitation-runoff relationships).
>
> Changes: we proposed a revised title.

**Did authors focus on the effect of drought on shifts in precipitation-runoff relationships and the performance of the model? Authors need to clarify the catchment/s studies. Or they use one catchment with sub-basins, and if so, the results reported should be for the basins?? The abstract has to be summarized and comprehensive.**

> We focused on three points: (1) we quantified shifts in precipitation-runoff relationships in four nested catchments of the Feather River; (2) we assessed performances of the PRMS model in these nested catchments during droughts and in particular during periods corresponding to shifts in the water balance; (3) by leveraging the fact that the performance of the model was sensitive to these shifts, we identified the water-balance term for which accuracy during droughts was statistically different from accuracy during wet periods ($ET$). We concluded that a different response time of $ET$ to precipitation variability is the likely cause of these shifts. We called this climate elasticity of $ET$. This point is general and goes beyond the specific catchments we considered.
>
> Changes: we revised the Abstract to clarify these points.

**Introduction**

**The motivation and novelty of the study is very week. The introduction lacks coherence. For example, the reader has to be able to identify:**

**1. overall effect of droughts on water balance in Med climate, with specific examples.**

**2. The approaches of examining precipitation-runoff relationships and how successful they have been in the same climate.**

**3. What has been done so far in relation to precipitation-runoff relationships or water balance studies within the same climate. This indicate the contribution of the work and its novelty.**

> Changes: we completely revised the Introduction following the structure recommended by the Referee. We also rearranged the sub-sections in Discussion to better align with the

Introduction.

**Research questions: Authors need to improve and rewrite all their research questions to be clear. It's very difficult for the reader to understand them. They raise a lot of questions: what is the water-balance predictive skill? Are the authors referring to the potential of the model to predict shifts during drought and non-drought periods???**

Changes: we worked on improving clarity of our research questions.

**What is the suitability of the generalized additive model in predicting ET for the catchment? Was it used before?**

The procedure is a modification of the well-cited approach by [5], which has been used in multiple papers since then [6, 3, 1, 7]. The main change compared to [5] is the addition of precipitation as predictor, which both recognizes the responsiveness of *ET* to precipitation, independent of NDVI, and provides a lower error. Further details can be found in [12].

Changes: we added further details about the procedure used to estimate ET in the main text **(see lines 1ff page 6)**, and also point the reader to the recently published paper by [12] – see an author formatted, accepted version at `https://escholarship.org/uc/item/4kb947md`.

**What were the characteristics of the Landsat-based annual-averaged NDVI? How was it derived? Is it a product or authors derived their own product through estimation? Any preprocessing of the image?**

Landsat 5, 7 and 8 were used to map NDVI at 30-m resolution. Values were calculated from the Tier 1 surface-reflectance product downloaded from Google Earth Engine. NDVI values among different Landsat sensors were homogenized by cross-calibrating Landsat 7 (NDVI in 2012) and Landsat 8 (NDVI in 2013-2016) into Landsat 5. Annual Landsat NDVI maps were generated by averaging all pixels in a water year. Pixels with shadow, snow, or cloud were excluded from the calculation.

Changes: we thought this too much detail for the manuscript and point the reader to the recently published paper by [12] for firther details – see an author formatted, accepted version at `https://escholarship.org/uc/item/4kb947md`.

**I suggest authors use their main objectives as subheadings for the reader to understand the findings and their implications.**

The three subsections of the Discussion do not directly relate to the main objectives stated in the Introduction, but look at results from other perspectives to further clarify their implications.

Changes: the main results of the paper were clarified in the first paragraph of the discussion.

[revised manuscript text omitted]

---

## Author Response (AR2)

Savona (Italy)

June 25, 2020

Dear Editor,

We would like to submit the manuscript *Climate elasticity of evapotranspiration shifts the water balance of Mediterranean climates during multi-year droughts* for publication in Hydrology and Earth System Sciences. This is a resubmission of manuscript **hess-2019-377**.

We thank the Editor for his fast response and helpful comments. We have revised and improved the manuscript accordingly, including both comments in the uploaded pdf and those in the formal review.

Please find attached our point-by-point replies and the new version of our manuscript for details. We also attached a version of the manuscript with tracked changes.

*Francesco Avanzi and coauthors*

**Reply to the Editor**

This point-by-point reply merges comments in the formal review and those in the annotated pdf, since the former reads as a summary of the most important points raised in the latter. We did not report minor comments on word choice, which were all addressed in the revised manuscript.

Comments regarding the Title

**The current title "Climate elasticity of evapotranspiration during multi-year droughts shifts the water balance of Mediterranean rain-snow climates" is very confusing. Is the core-issue being investigated "climate elasticity of evapotranspiration" or the influence of the depletion of subsurface water on contributions of the various elements of the water balance". One of the Reviewers highlighted the problem of the title.**

> This manuscript seeks process-based explanations for shifts in the precipitation-runoff relationship, that is, the fundamental relationship between annual precipitation and annual runoff. While shifts in this relationship during droughts have been reported in Australia [6] and China [8], such explanations are still lacking.
> We found that these shifts are the result of a hysteretic response of the water budget to multi-year droughts. The driver is a delayed response of evapotranspiration to precipitation interannual variability, a mechanism that we defined as climate elasticity of evapotranspiration. The depletion of subsurface water storage characterizes the early stages of this hysteresis response, when evapotranspiration in excess to precipitation is supported by sub-surface storage.
> So the manuscript primarily focuses on shifts in the precipitation-runoff relationship, with climate elasticity of evapotranspiration and the influence of the depletion of subsurface water on contributions of the various elements of the water balance being process-based explanations for these shifts.

**I fail to understand what the authors mean by "multi-year droughts shifts". I suspect the authors wanted to say how multi-year droughts affect the elements of the water balance. I am not sure whether this can be considered a shift, but just a response to reduced carryover storage.**

> "Multi-year droughts shifts" was not intended as a unique group of words, since "shifts" was the verb of our declarative title. This was certainly unclear.

**The tem "Mediterranean rain-snow climates" will just confuse readers. Why not state simply "Mediterranean climate".**

> Agreed

> Changes to all comments above: we revised the title in view of all these comments. The new title reads *Climate elasticity of evapotranspiration shifts the water balance of Mediterranean climates during multi-year droughts*, which should both highlight the main topic of the paper (shifts in the water balance during droughts) and avoid issues raised above.

Other comments

**Page 2, Line 16. The authors should avoid referring to transformation of rainfall into surface runoff or subsurface runoff as an allocation or "priority allocation". The word allocation is generally understood in water resources management to refer to granting a user a portion of the water.**

The word "allocation" was first used by [1] and certainly had a suggestive meaning. Nevertheless, we agree that this word may read confusing.

Changes: we replaced "allocation" with "partitioning" throughout the paper. The word "partitioning" was used with this meaning by [6] in a seminal paper about shifts in rainfall-runoff relationship in Australia.

**But the title refers to rain-snow climates?**

Correct. The first sentence of the abstract was aimed to introduce previous findings in this field, which all focus on rainfall-dominated regions rather than mixed rain-snow regions.

Changes: we removed the reference to rainfall-dominated climates in the Abstract and the reference to mixed rain-snow regions from the Title to make our statements more general and clear.

**what do you mean by "fundamental implications"?**

We meant that understanding what causes shifts in the precipitation-runoff relationship is important for both basic and applied science.

Changes: we removed these words, since they were unnecessary and unclear.

**Simplify this. One interpretation is that you are referring to proportions of rain and snow contributing to total precipitation**

Changes: Agreed. We removed these words from that line of the Abstract and from the Title. Editor's interpretation is correct: this word is used to denote that our study region is neither rainfall- nor snowfall-dominated, but both snowfall and rainfall contribute to annual precipitation

**It is not good practice to use basin and catchment in the same paper. Stick to one of them throughout.**

Changes: We replaced catchment with basin throughout the paper.

**It is not clear what the statistical significance is referring to. Is this the change of ET or predictive skill of PRMS?**

Changes: It is the predictive skill of PRMS. This was clarified.

**what do you mean by buffered response?**

We mean that the response of $ET$ to precipitation variability is delayed.

Changes: This was clarified.

**In some catchments storage combines both surface and sub-surface components, e.g. where there are lakes or wetlands.**

> Correct. We assume here that the contribution of lakes and wetlands to $\Delta S$ is negligible.
> Changes: Clarified.

**why do you say "ample"? Ample with respect to what?**

> Changes: this was replaced with "ample water for sustaining dry-season $ET$", which clarified that "ample" is with respect to dry-season $ET$ demand.

**Is the low interannual variability due to low water storage as this sentence seems to imply? If so be explicit.**

> No, low interannual variability was despite low water storage.
> Changes: This was clarified.

**You need to be specific. a plausible interpretation is you are saying types of rocks affect the P-ET relationship**

> Changes: we repllaced "geology" with "regolith lithology", which may intensify or alleviate the impact of changing $P$ on $Q$ in Mediterranean climates (the main subject of this paragraph, see lines 12ff page 2) through subsurface storage and groundwater flow.

**Cannot be valid for physically based models. Yes for models calibrated during non-drought periods, this is expected.**

> [2] and [5] focused on conceptual hydrologic models, and so we added "conceptual" in this sentence. Nevertheless, this paper shows that even a physically based model calibrated during drought periods (PRMS) is exposed to a drop in predictive accuracy during droughts. This points to the cause of this drop in predictive accuracy being a process that the model does not consider, which is usually referred to as conceptual uncertainty [10]. We demonstrate that the source of conceptual uncertainty during droughts is climate elasticity of evapotranspiration.

**Are you not really referring to groundwater influence on streamflows?**

> We refer to the predominant basin lithology and so predominance of surface runoff over groundwater baseflow. This was clarified.

**You may need to be explicit about the types of models you are referring to. If a model does realistically represent the physical processes, then there should not be any changes in performance due to changes in inputs and outputs (precip and ET)**

> See our reply above about conceptual uncertainty.

**Please state the criterion or criteria used to define a drought in California. Readers need to know this without having to go to the website you are citing. The definition of a drought is important in understanding this paper.**

The designation of drought years in California focuses on impacts to water users, and so these water years correspond to a hydrologic- and socioeconomic- rather than a meteorological- or agricultural-drought condition [see definitions in 9]. In practice, this designation is based on a broad array of real-time indeces collected throughout the State, including precipitation anomalies, snowpack accumulation, forecasted water supply for the dry season, and drought levels according to the U.S. Drought Monitor at `https://droughtmonitor.unl.edu/` (visited June 13, 2020).

Changes: This information was added to the paper (**see lines 13ff page 4**).

**Another expression used is "full-natural flow", what does this mean? Is there any half-full natural flow?**

**Page 6, Line 16-17. Please describe the method used by Pacific Gas & Electric (PG&E) to naturalize daily flows. Readers have no idea how this was done, and the reliability of the method used.**

Full-natural flow – sometimes also referred to as unimpaired flow or reconstructed streamflow – is a mass-balance reconstruction of runoff as if no dam or other man-made infrastructure affected it [a rare condition in California, see 3]. This reconstruction is achieved using published gauged streamflow and reservoir gauge data, as well as estimates of unmeasured inflows and/or outflows. Thus, computing full-natural flow requires extensive knowledge of water infrastructures and reservoir operations.

Note that full-natural flow rather than natural flow is used because the latter may convey the false idea that data refer to actually observed flow conditions. Using "natural flow" *in lieu* of full-natural flow is thus discouraged by the California Water Boards (`https://www.waterboards.ca.gov/waterrights/water_issues/programs/bay_delta/sds_srjf/sjr/docs/dwr_uf010611.pdf`), whereas our experience is that full-natural flow is a frequently used term in hydrologic practice of the western US.

Changes: This information was added to the paper (**see lines 23ff page 5**).

**Is this not speculation since you did not investigate the accessible of sub-surface water to plants?**

We agree that this may be to a speculation to some extent, but also think that the reader may benefit from a process-based explanation of the observed bias. The use of "could" clarifies that this is just an hypothesis.

**Is this not a reflection that the model does not realistically represent some of the processes?**

Correct. This was made even clearer.

**The influence of carryover of soil water on Q and ET is valid for all catchments. Avoid giving an impression that this is unique for mediterranean catchments. This behaviour has been observed even in grasslands, and wetlands**

Agreed and fixed.

**Is this just due to poor representation of soil water and groundwater?**

We believe this is due to the model likely miscapturing the magnitude and distribution of regolith moisture. This was clarified.

**Page 15, Line 15, "These four phases associated with precipitation-runoff hysteresis have never been investigated in a systematic way". My understanding is that the influence of the subsurface water on runoff generation before, during and after a drought is similar to any storage in a catchment. This could be in surface storage such as in wetlands. One of the Reviewers highlighted that the lack of adequate representation of groundwater in PRMS contributes to poor representation of runoff generation during the post-drought period. The poor performance of conceptual models after droughts is a clear reflection that the models are not adequately representing the physical process.**

The main point of this paper is that the driver of increased runoff anomaly during droughts compared to non-drought periods is the *ET* response rather than groundwater. Note also that at least one of the sub-basin considered in this paper is surface-runoff-dominated, and this dominance of surface runoff is a common feature of most Californian Sierra-Nevada catchments due to the prevalence of granitic outcrops. So we expect the lack of adequate representation of groundwater in PRMS to play a minor role in this analysis.

Changes: this passage was revised for clarity.

**There are several studies that have examined how droughts influence runoff generation and the relative importance of components of the water balance. See for example these references:**

**Lange, J. and Haensler, A., 2012. Runoff generation following a prolonged dry period. Journal of Hydrology, 464, pp.157-164.**

**Saft, M., Western, A.W., Zhang, L., Peel, M.C. and Potter, N.J., 2015. The influence of multiyear drought on the annual rainfall-runoff relationship: An Australian perspective. Water Resources Research, 51(4), pp.2444-2463.**

We agree. Actually, [7] was already extensively cited in the manuscript and represented an important reference and source of inspiration for this research. [4] is now cited as well.

[revised manuscript text omitted]